# Mending synthetic data with MAPS: Model Agnostic Post-hoc Synthetic Data Refinement Framework

## Abstract

Generating high-quality synthetic data with privacy protections remains a challenging ad-hoc process, requiring careful model design and training often tailored to the characteristics of a targeted dataset. We present MAPS, a model-agnostic post-hoc framework that improves synthetic data quality for any pre-trained generative model while ensuring sample-level privacy standards are met. Our two-stage approach first removes synthetic samples that violate privacy by being too close to real data, achieving 0-identifiability guarantees. Second, we employ importance weighting via a binary classifier to resample the remaining synthetic data according to estimated density ratios. We evaluate MAPS across two healthcare datasets (TCGA-metadata, GOSSIS-1-eICU-cardiovascular) and four generative models (TVAE, CTGAN, TabDiffusion, DGD), demonstrating significant improvements in fidelity and utility while maintaining privacy. Notably, MAPS achieves substantial improvements in fidelity metrics, with 40 out of 48 statistical tests demonstrating significant improvements in marginal distributional measures and notable enhancements in correlation structure preservation and joint distribution similarity. For example, Joint Jensen-Shannon Distance reduced from ranges of 0.7888-0.8278 to 0.5434-0.5961 on TCGA-metadata and 0.6192-0.7902 to 0.3633-0.4503 on GOSSIS-1-eICU-cardiovascular. Utility improvements are equally impressive, with classification F1 scores improving from ranges of 0.0866-0.2400 to 0.3043-0.3848 on TCGA-metadata and 0.1287-0.2085 to 0.2104-0.2497 on GOSSIS-1-eICU-cardiovascular across different model-dataset combinations. Additionally, uncertainty quantification analysis via split conformal prediction demonstrates that MAPS considerably improves calibration quality, reducing average prediction set sizes by 55-77% while maintaining target coverage on TCGA-metadata. The code of this project is available at `https://anonymous.4open.science/r/MAPS-EBF8`.

## 1 Introduction

The proliferation of data-driven applications in privacy-sensitive domains has created an urgent need for synthetic data that preserves statistical fidelity while protecting individual privacy (van Breugel et al., 2024; Tucker et al., 2020). However, generating high quality synthetic data that satisfies both privacy requirements and downstream task performance remains challenging, requiring practitioners to navigate complex trade-offs between competing objectives - privacy and fidelity (Kaabachi et al., 2025; Yan et al., 2022). This privacy-fidelity trade-off forces practitioners to choose between models that provide strong privacy guarantees but produce data of limited practical value, and models that generate high fidelity synthetic data but potentially leak sensitive information. This challenge is compounded by the substantial technical expertise required to design, train, and tune generative models effectively (Belgodere et al., 2024; Padariya et al., 2025).

Existing approaches typically fall into two distinct categories, each with significant limitations. Privacy-first methods such as ADS-GAN (Yoon et al., 2020), PATE-GAN (Jordon et al., 2018), and DP-GAN (Xie et al., 2018) incorporate privacy constraints during training, providing formal privacy guarantees but often producing synthetic data with notably degraded utility. Conversely, fidelity-first approaches like TVAE (Xu et al., 2019), CTGAN (Xu et al., 2019), TabDDPM (Kotelnikov et al.,

2023) and DGD (Schuster & Krogh, 2023) focus on generating high fidelity synthetic data but no privacy protections beyond sampling design.

We present MAPS (Model Agnostic Post-hoc Synthetic Data Refinement Framework), a novel two-stage approach to refining synthetic datasets from any generator that addresses both privacy and fidelity concerns through complementary mechanisms. The first stage minimizes re-identification risks by implementing 0-identifiability guarantees, systematically removing synthetic samples that violate privacy by being closer to real samples than those real samples are to their nearest real neighbors. The second stage enhances data fidelity through theoretically grounded importance weighting, where we train a binary classifier to distinguish between real and synthetic data, then use this classifier to estimate density ratios following the likelihood-free importance weighting framework (Grover et al., 2019). The framework is currently designed to refine static tabular synthetic data produced by any generation method.

## 2 METHODOLOGY

In this section, we first formalize the problem setup, then detail the privacy filtering mechanism that enforces 0-identifiability constraints, describe the importance weighting approach that improves distributional fidelity through density ratio estimation, and finally present the Sampling-Importance-Resampling procedure for selecting the refined synthetic dataset.

### 2.1 PROBLEM FORMULATION

Let $\mathcal{D} = \{x_i\}_{i=1}^N$ denote a dataset of $N$ real samples drawn i.i.d. from an unknown distribution $p(x)$, and $\hat{\mathcal{D}} = \{\hat{x}_j\}_{j=1}^M$ denote a synthetic dataset of $M$ samples generated by some generative model with distribution $p_\theta(x)$. Our objective is to refine $\hat{\mathcal{D}}$ to produce a subset $\tilde{\mathcal{D}} \subset \hat{\mathcal{D}}$ of size $N$ that (1) provides formal identifiability protections with respect to $\mathcal{D}$, and (2) exhibits improved fidelity to the true data distribution $p(x)$.

### 2.2 STAGE 1: PRIVACY FILTERING

The first stage protects privacy by removing synthetic samples that violate identifiability constraints. We build upon the $\epsilon$-identifiability framework (Yoon et al., 2020) and set it to be 0-identifiability to maximize protection using this privacy standard. Note that we can use any privacy metric that works on single samples in this stage.

For each real sample $x_i \in \mathcal{D}$, we define its distinctness threshold as:

$$r_i = \min_{x_j \in \mathcal{D} \setminus \{x_i\}} \|\mathbf{w} \cdot (x_i - x_j)\| \tag{1}$$

where $\mathbf{w}$ is a feature weight vector that accounts for the relative importance of different features in measuring similarity[1].

For each real sample $x_i$, we also compute its proximity to the synthetic dataset:

$$\hat{r}_i = \min_{\hat{x}_j \in \hat{\mathcal{D}}} \|\mathbf{w} \cdot (x_i - \hat{x}_j)\| \tag{2}$$

The $\epsilon$-identifiability of synthetic dataset $\hat{\mathcal{D}}$ with respect to real dataset $\mathcal{D}$ requires that:

$$\mathcal{I}(\mathcal{D}, \hat{\mathcal{D}}) = \frac{1}{N} \sum_{i=1}^N \mathbb{I}(\hat{r}_i < r_i) < \epsilon \tag{3}$$

where $\mathbb{I}$ is the indicator function.

In MAPS, we enforce the constraint of 0-identifiability by removing all synthetic samples $\hat{x}_j$ such that there exists a real sample $x_i$ for which $\|\mathbf{w} \cdot (x_i - \hat{x}_j)\| < r_i$. This ensures that no synthetic sample is closer to any real sample than that real sample's nearest real neighbor, providing the strongest possible identifiability guarantee using this measure.

---

[1]In our implementations, we treat all features with the same weight 1.

## 2.3 STAGE 2: FIDELITY ENHANCEMENT VIA IMPORTANCE WEIGHTING

The second stage improves the fidelity of the identifiability-filtered synthetic data through importance weighting and resampling. We train a binary probabilistic classifier $c_\phi(x) : \mathcal{X} \to [0, 1]$ to distinguish between real and synthetic samples, then use this classifier to estimate importance weights.

The classifier outputs an estimate of the probability that a given sample belongs to the real data distribution:

$$c_\phi(x) = P(y = 1|x) \tag{4}$$

where $y = 1$ indicates that sample $x$ comes from the real data distribution $p(x)$.

Using Bayes' theorem, we can express this probability as:

$$c_\phi(x) = \frac{p(x)\pi_1}{p(x)\pi_1 + p_\theta(x)\pi_0} \tag{5}$$

where $\pi_1 = P(y = 1)$ and $\pi_0 = P(y = 0)$ are the prior probabilities of observing samples from the real and synthetic distributions, respectively, in the training set used for the classifier.

The estimated importance weight, representing the estimated density ratio $\frac{p(x)}{p_\theta(x)}$, can be derived as:

$$\hat{w}_\phi(x) = \frac{p(x)}{p_\theta(x)} = \frac{\pi_0}{\pi_1} \frac{c_\phi(x)}{1 - c_\phi(x)} \tag{6}$$

This formulation follows the ideas from the likelihood-free importance weighting framework of (Grover et al., 2019), enabling us to estimate density ratios without explicit density models.

## 2.4 SAMPLING-IMPORTANCE-RESAMPLING

Using the estimated importance weights, we apply Sampling-Importance-Resampling (SIR) to select a refined synthetic dataset. Given importance weights for the $M'$ identifiability-filtered synthetic samples, we normalize these weights and sample $N$ samples according to the normalized probabilities:

$$p_j = \frac{\hat{w}_\phi(\hat{x}_j)}{\sum_{N=1}^{M'} \hat{w}_\phi(\hat{x}_N)} \tag{7}$$

The final refined synthetic dataset $\tilde{\mathcal{D}}$ contains $N$ samples that satisfy both identifiability constraints and exhibit improved fidelity to the real data distribution. The full algorithm can be found in Appendix A. Figure 1 provides an overview of our two-stage approach to achieve these objectives.

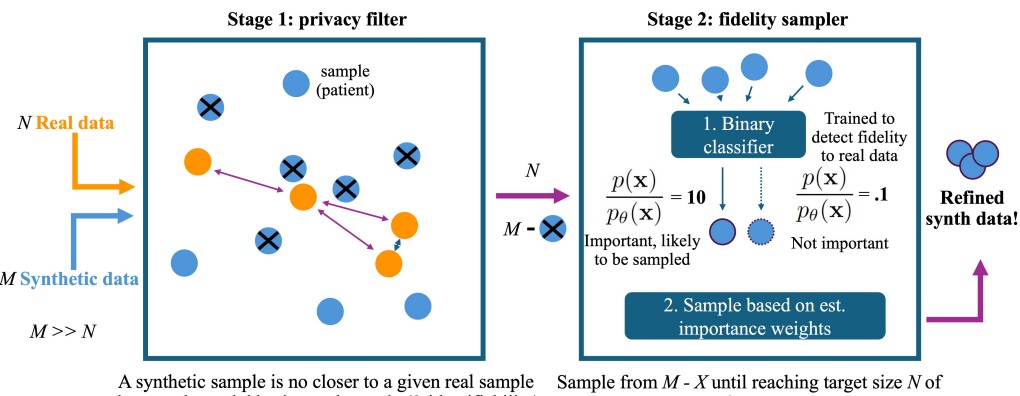

Figure 1: Overview of the MAPS two-stage framework. Stage 1 removes synthetic samples violating 0-identifiability (marked with ×, with the number of $X$). Stage 2 trains a binary classifier to estimate importance weights via density ratios, then applies SIR to produce refined synthetic data that better approximates the real data distribution.

# 3 EXPERIMENTAL SETUP

**Generative Models.** We investigate MAPS' capabilities using four representative generative models covering the major paradigms in tabular data synthesis:

- **TVAE** (Xu et al., 2019): A representative variational autoencoder specifically designed for tabular data generation.
- **CTGAN** (Xu et al., 2019): The most widely adopted GAN-based model for tabular synthetic data generation.
- **TabDDPM** (Kotelnikov et al., 2023): A diffusion-based model representing the latest paradigm in generative modeling for tabular data.
- **DGD** (Schuster & Krogh, 2023): An encoder-free deep generative decoder with simple architecture that enhances learning of multi-modal data.

**Datasets.** We evaluate on two publicly available healthcare datasets:

- **TCGA-metadata**: The largest public cancer dataset covering 33 different cancer types with 11,315 records. We selected 21 variables with minimal missing data for our experiments.
- **GOSSIS-1-eICU-cardiovascular**: A large public cardiovascular ICU dataset with 41,396 records and 68 variables, representing complex clinical data with mixed data types.

The detailed data pre-processing procedure is described in Appendix B.1.

**Evaluation Protocol.** For each generative model and dataset combination, we generate a synthetic data pool of size $M = 30N$ where $N$ is the number of real samples. This large pool enables meaningful selection during the refinement process. The baseline synthetic dataset consists of $N$ randomly sampled synthetic samples from this pool, while the MAPS-refined dataset also contains $N$ samples but selected through our two-stage refinement process. We assess MAPS effectiveness across three dimensions: (1) distributional fidelity using statistical tests and similarity measures to evaluate how well synthetic data captures real data distributions, (2) utility preservation through downstream task performance including clustering and classification tasks(with uncertainty quantification evaluation) using a "train-on-synthetic, test-on-real" evaluation scheme, and (3) privacy protection via resistance to membership inference attacks to ensure refinement does not compromise privacy standards. A detailed description of the implementations and evaluation methods can be found in Appendix B and Appendix C respectively.

# 4 RESULTS AND DISCUSSION

In this section, we present a comprehensive evaluation of MAPS across multiple dimensions to demonstrate its effectiveness in improving synthetic data quality while maintaining privacy protections. Our analysis examines distributional fidelity improvements, utility enhancements in downstream tasks, and privacy preservation across two healthcare datasets and four generative models.

## 4.1 MARGINAL DISTRIBUTION SIMILARITY

As an illustrative example of improvements in distributional alignment, we compare selected marginal distributions between the real data, the raw synthetic data, and the refined synthetic data produced by MAPS. Figure 2 shows examples across numerical and categorical variables demonstrating the benefits of MAPS refinement. For instance, when examining the `initial_weight` variable for TVAE, the raw synthetic data exhibits spurious fluctuations when `initial_weight` is around 100, which are eliminated after refinement. Similarly, for TabDDPM, the problematic bump in the long tail region of the raw synthetic data distribution is corrected in the refined version. Categorical variable improvements manifest as refined synthetic data better reflecting the true proportional distributions of real data categories, with more accurate frequency representations across all category levels.

The quantitative assessment in Table 1 reveals that 40 out of 48 statistical tests demonstrate significant improvements in marginal distributional measures. The Jensen-Shannon Distance shows particularly impressive improvements across all model-dataset combinations: TVAE demonstrates substantial

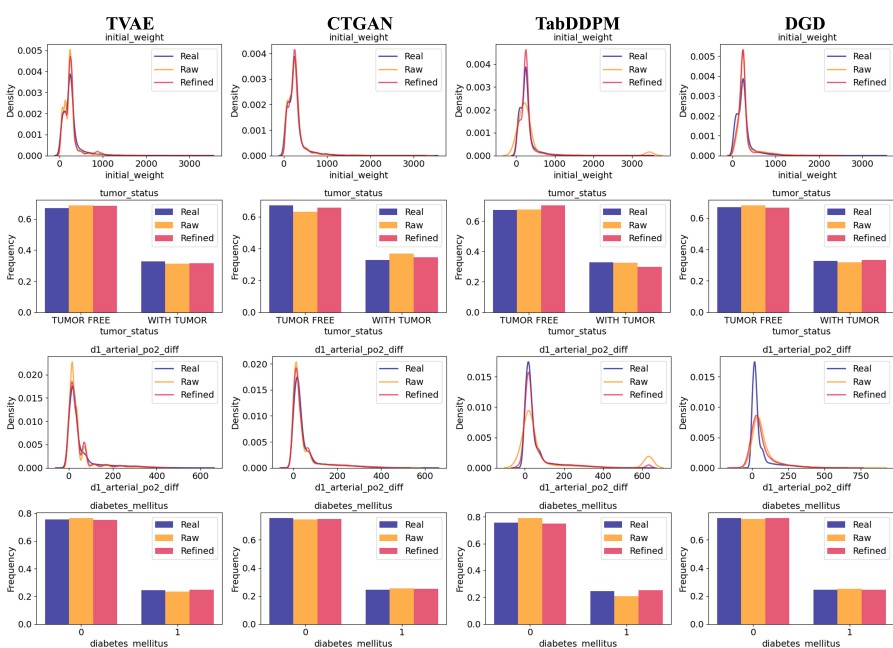

Figure 2: Marginal distribution comparison across models and variables. The 4×4 subplots compare marginal distributions across generative models and variables, including: `initial_weight` and `tumor_status` from TCGA metadata (numerical and categorical, respectively), and `d1_arterial_po2_diff` and `diabetes_mellitus` from GOSSIS-1-eICU-cardiovascular. Real refers to the real data, Raw to raw synthetic data, and Refined to MAPS-refined outputs.

reductions from 0.0807 to 0.0388 on TCGA-metadata and from 0.0652 to 0.0308 on GOSSIS-1-eICU-cardiovascular, while CTGAN achieves similar dramatic decreases from 0.0760 to 0.0373 on TCGA-metadata and from 0.0443 to 0.0269 on GOSSIS-1-eICU-cardiovascular. Total Variation Distance exhibits equally striking improvements, with TVAE reducing from 0.0684 to 0.0312 on TCGA-metadata and CTGAN dropping from 0.0427 to 0.0222 on GOSSIS-1-eICU-cardiovascular.

Table 1: Marginal distribution fidelity assessment. Results show mean ± standard deviation across multiple runs. Arrows indicate improvement direction: ↑ higher is better, ↓ lower is better. Bold values denote statistically significant improvements ($p < 0.05$, paired $t$-test) between raw and refined results for each model. This criterion is used consistently throughout all result tables.

**TCGA-metadata**

| Metric | TVAE | | CTGAN | | TabDDPM | | DGD | |
|---|---|---|---|---|---|---|---|---|
| | Raw | Refined | Raw | Refined | Raw | Refined | Raw | Refined |
| Kolmogorov-Smirnov ↑ | 0.8142±0.0017 | **0.8180**±0.0008 | 0.8107±0.0010 | 0.8110±0.0015 | 0.9433±0.0009 | **0.9492**±0.0029 | 0.9082±0.0011 | **0.9439**±0.0013 |
| Chi-square Test ↑ | 0.5422±0.1286 | 0.6130±0.0406 | 0.8738±0.0007 | 0.7518±0.1017 | 0.7182±0.0914 | 0.6775±0.0500 | 0.7222±0.0008 | 0.7225±0.0006 |
| Jensen-Shannon Dist. ↓ | 0.0807±0.0008 | **0.0388**±0.0012 | 0.0760±0.0006 | **0.0373**±0.0021 | 0.0479±0.0012 | **0.0347**±0.0013 | 0.0402±0.0005 | **0.0362**±0.0008 |
| Total Variation Dist. ↓ | 0.0684±0.0010 | **0.0312**±0.0015 | 0.0734±0.0005 | **0.0328**±0.0029 | 0.0356±0.0012 | **0.0263**±0.0013 | 0.0391±0.0006 | **0.0318**±0.0011 |
| Hellinger Distance ↓ | 0.0845±0.0009 | **0.0397**±0.0012 | 0.0780±0.0006 | **0.0377**±0.0021 | 0.0499±0.0013 | **0.0354**±0.0013 | 0.0406±0.0005 | **0.0366**±0.0008 |
| Inverse KL Divergence ↑ | 0.9345±0.0026 | **0.9808**±0.0007 | 0.9604±0.0005 | **0.9883**±0.0005 | 0.9821±0.0013 | **0.9901**±0.0024 | 0.9860±0.0005 | **0.9910**±0.0004 |

**GOSSIS-1-eICU-cardiovascular**

| Metric | TVAE | | CTGAN | | TabDDPM | | DGD | |
|---|---|---|---|---|---|---|---|---|
| | Raw | Refined | Raw | Refined | Raw | Refined | Raw | Refined |
| Kolmogorov-Smirnov ↑ | 0.8556±0.0003 | **0.8738**±0.0004 | 0.8631±0.0004 | **0.8788**±0.0003 | 0.8746±0.0016 | **0.9707**±0.0003 | 0.8067±0.0008 | **0.8352**±0.0006 |
| Chi-square Statistic ↑ | 0.8061±0.0021 | **0.8839**±0.0366 | 0.9061±0.0302 | 0.9201±0.0006 | 0.8096±0.0016 | **0.8357**±0.0009 | 0.8767±0.0283 | 0.9022±0.0007 |
| Jensen-Shannon Dist. ↓ | 0.0652±0.0005 | **0.0308**±0.0002 | 0.0443±0.0008 | **0.0269**±0.0005 | 0.0665±0.0007 | **0.0418**±0.0005 | 0.0472±0.0004 | **0.0406**±0.0005 |
| Total Variation Dist. ↓ | 0.0595±0.0003 | **0.0244**±0.0004 | 0.0427±0.0008 | **0.0222**±0.0003 | 0.0596±0.0006 | **0.0436**±0.0005 | 0.0408±0.0003 | 0.0438±0.0005 |
| Hellinger Distance ↓ | 0.0667±0.0005 | **0.0312**±0.0002 | 0.0455±0.0008 | **0.0272**±0.0006 | 0.0681±0.0007 | **0.0422**±0.0005 | 0.0493±0.0004 | **0.0411**±0.0005 |
| Inverse KL Divergence ↑ | 0.9601±0.0005 | **0.9867**±0.0009 | 0.9705±0.0012 | **0.9882**±0.0004 | 0.9694±0.0004 | **0.9811**±0.0003 | 0.9807±0.0003 | **0.9857**±0.0003 |

Inverse KL divergence is consistently enhanced, with values moving from the 0.93-0.98 range for raw synthetic data to above 0.98 for all refined models. Notably, CTGAN shows improvements from 0.9604 to 0.9883 on TCGA-metadata and from 0.9705 to 0.9882 on GOSSIS-1-eICU-cardiovascular, while TabDDPM achieves near-perfect scores improving from 0.9821 to 0.9901 on TCGA-metadata. These consistent improvements across distributional metrics highlight MAPS's effectiveness in

correcting marginal misalignments across diverse generative paradigms. Moreover, while our results reveal variability in output quality among generative models, they also underscore the practical value of MAPS as a flexible post-hoc add-on applicable to any architecture or training paradigm.

## 4.2 JOINT DISTRIBUTION AND CORRELATION STRUCTURE

Joint distribution and correlation structure improvements via synthetic data refinement with MAPS are equally impressive.

Table 2: Joint distribution similarity and correlation structure preservation. WD = Wasserstein Distance (joint numerical distributions), JSD = Joint Jensen-Shannon Distance (joint categorical distributions), NFN = Normalized Frobenius Norm (correlation structure differences). Lower values indicate better fidelity.

| | **TCGA-metadata** | | | | | | | |
| --- | --- | --- | --- | --- | --- | --- | --- | --- |
| | **TVAE** | | **CTGAN** | | **TabDDPM** | | **DGD** | |
| Metric | Raw | Refined | Raw | Refined | Raw | Refined | Raw | Refined |
| WD ↓ | 0.0145±0.0004 | **0.0056**±0.0002 | 0.0130±0.0003 | **0.0073**±0.0003 | 0.1099±0.0048 | **0.0089**±0.0005 | 0.0291±0.0007 | **0.0102**±0.0002 |
| JSD ↓ | 0.7909±0.0041 | **0.5562**±0.0020 | 0.8278±0.0034 | **0.5736**±0.0028 | 0.8239±0.0029 | **0.5434**±0.0018 | 0.7888±0.0045 | **0.5961**±0.0040 |
| NFN ↓ | 0.0770±0.0014 | **0.0348**±0.0011 | 0.0749±0.0006 | **0.0376**±0.0020 | 0.0927±0.0019 | **0.0245**±0.0013 | 0.1330±0.0005 | **0.0421**±0.0009 |

| | **GOSSIS-1-eICU-cardiovascular** | | | | | | | |
| --- | --- | --- | --- | --- | --- | --- | --- | --- |
| | **TVAE** | | **CTGAN** | | **TabDDPM** | | **DGD** | |
| Metric | Raw | Refined | Raw | Refined | Raw | Refined | Raw | Refined |
| WD ↓ | 0.0993±0.0024 | **0.0921**±0.0003 | 0.1221±0.0014 | **0.1004**±0.0007 | 1.3132±0.0392 | **0.2418**±0.0122 | 0.3159±0.0061 | **0.1540**±0.0022 |
| JSD ↓ | 0.7547±0.0013 | **0.4187**±0.0014 | 0.7902±0.0022 | **0.4503**±0.0010 | 0.7241±0.0013 | **0.3633**±0.0009 | 0.6192±0.0016 | **0.4197**±0.0014 |
| NFN ↓ | 0.0546±0.0005 | **0.0364**±0.0012 | 0.0523±0.0004 | **0.0399**±0.0003 | 0.1190±0.0011 | **0.0342**±0.0004 | 0.0686±0.0004 | **0.0527**±0.0004 |

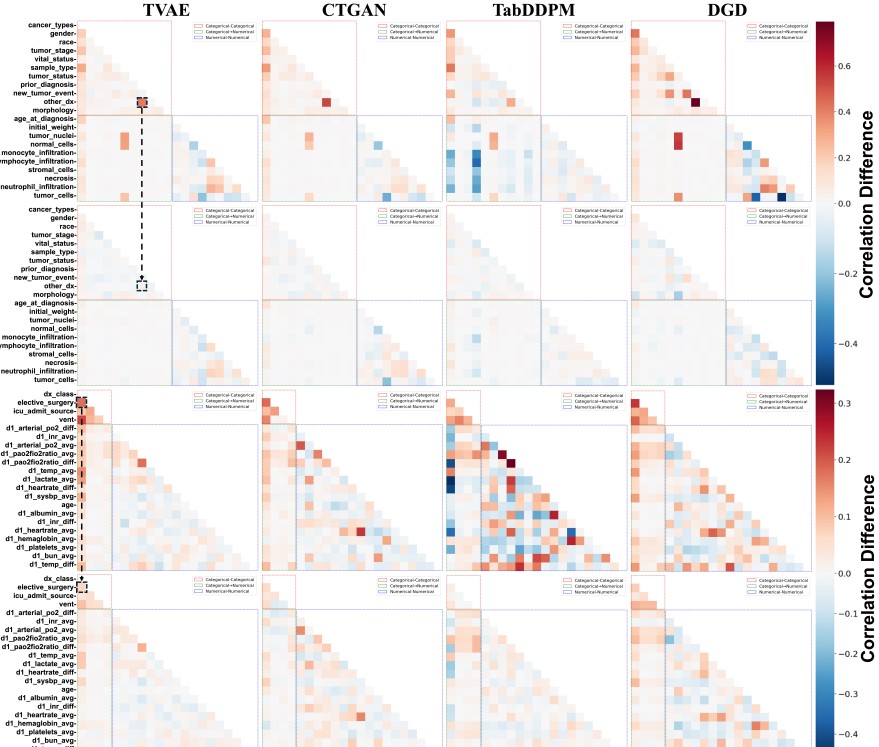

Figure 3: Correlation structure difference analysis. The figure shows 4×4 subplots where columns represent generative models. Rows 1-2 show TCGA-metadata correlation matrix differences (real vs. raw synthetic, real vs. refined synthetic), and rows 3-4 show GOSSIS-1-eICU-cardiovascular differences. Darker colors indicate larger differences from the real data correlation structure.

Table 2 shows significant improvements across all joint distribution metrics. The Joint Jensen-Shannon Distance is substantially reduced for all model-dataset pairs: on TCGA-metadata, improvements range from 24.4% (DGD: 0.7888→0.5961) to 34.0% (TabDDPM: 0.8239→0.5434). On GOSSIS-1-eICU-cardiovascular, even larger gains are observed: TVAE 44.5% (0.7547→0.4187), CTGAN 43.0% (0.7902→0.4503), TabDDPM 49.8% (0.7241→0.3633), and DGD 32.2% (0.6192→0.4197). Correlation structure preservation, measured by normalized Frobenius norm, also improves universally, with reductions from 23.2% to 73.6%. Figure 3 illustrates these reductions, showing MAPS refinement aligns synthetic data correlations much closer to real patterns. On TCGA-metadata, raw synthetic data fail to capture the correlation between `other_dx` and `prior_diagnosis`, while refined data achieves near-perfect preservation. On GOSSIS-eICU-cardiovascular, correlations between `elective_surgery` and `dx_class` are weak in raw data but considerably improved after refinement.

## 4.3 UTILITY ENHANCEMENT RESULTS

The utility improvements observed across clustering and classification tasks directly address one of the most critical concerns for practitioners (Qian et al., 2024; Yoon et al., 2023). Figure 4 demon-

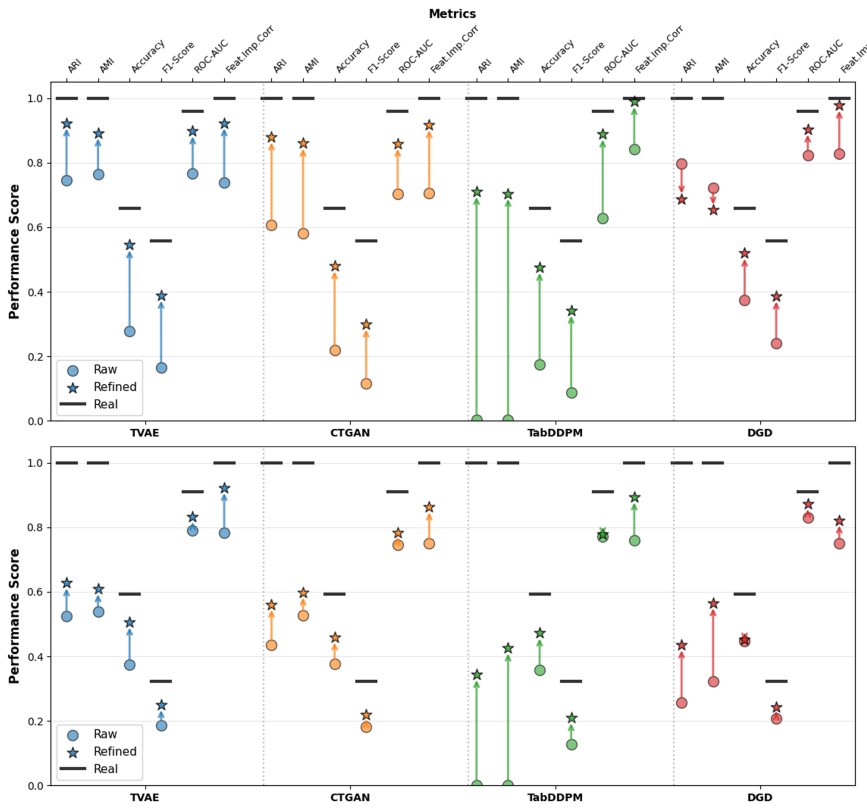

Figure 4: Downstream task performance comparison. The figure shows performance improvements after MAPS refinement across multiple tasks and metrics. Circles represent raw synthetic data performance, stars represent refined synthetic data performance, and the black horizontal line indicates the oracle performance (train on real data, test on real data). The upper and lower panels show results for TCGA-metadata and GOSSIS-1-eICU-cardiovascular datasets, respectively.

strates MAPS's ability to rescue low utility synthetic data—particularly TabDDPM's transformation from near-zero clustering performance (ARI: 0.0022) to strong clustering agreement (ARI: 0.7384) on TCGA-metadata. Classification task improvements are equally significant, with F1 scores improving from ranges of 0.0866-0.2400 to 0.3043-0.3848 on TCGA-metadata and from 0.1287-0.2085 to 0.2104-0.2497 on GOSSIS-1-eICU-cardiovascular across different model-dataset combinations. Uncertainty quantification (UQ) via split conformal prediction reveals critical quality differences be-

tween synthetic and real data, emphasizing the need for assessing UQ capability alongside traditional utility metrics to ensure robust AI systems. Table 3 shows that raw synthetic data often produces unusable uncertainty estimates: on TCGA-metadata, CTGAN, TabDDPM, and DGD achieve full coverage (100%, meaning all true labels fall within their prediction sets) but must include all 33 cancer types in prediction sets to do so, exceeding the required 95% target coverage and rendering the predictions uninformative, while TVAE provides imperfect coverage (89%) despite also using impractically large sets (17.2216). MAPS refinement brings coverage closer to target (0.9072-0.9626) while dramatically reducing set sizes to be 2.2-4.4 times smaller compared to raw synthetic data. On GOSSIS-1-eICU-cardiovascular, we observe coverage-set size tradeoffs, though DGD demonstrates simultaneous improvements in both coverage and average prediction set size. These results underscore that synthetic data evaluation should extend beyond prediction accuracy to encompass uncertainty quantification, as MAPS enhances both prediction performance and UQ reliability.

Table 3: Uncertainty quantification analysis using split conformal prediction with inverse probability score function ($\alpha = 0.05$, target coverage = 0.95) across synthetic data types.

| **TCGA-metadata** | | | | | | | | | |
|---|---|---|---|---|---|---|---|---|---|
| | **TVAE** | | **CTGAN** | | **TabDDPM** | | **DGD** | | **Oracle** |
| Metric | Raw | Refined | Raw | Refined | Raw | Refined | Raw | Refined | Real |
| Coverage | $0.8897 \pm 0.0037$ | $\mathbf{0.9072} \pm 0.0051$ | $1.0000 \pm 0.0000$ | $\mathbf{0.9243} \pm 0.0098$ | $1.0000 \pm 0.0000$ | $\mathbf{0.9626} \pm 0.0143$ | $1.0000 \pm 0.0000$ | $\mathbf{0.9269} \pm 0.0060$ | $0.9600 \pm 0.0043$ |
| Avg. Set Size | $17.2216 \pm 0.1972$ | $\mathbf{7.7150} \pm 0.0883$ | $33.0000 \pm 0.0000$ | $\mathbf{10.4542} \pm 1.1069$ | $33.0000 \pm 0.0000$ | $\mathbf{13.7615} \pm 1.7176$ | $33.0000 \pm 0.0000$ | $\mathbf{7.5504} \pm 0.6235$ | $5.5915 \pm 0.5476$ |

| **GOSSIS-1-eICU-cardiovascular** | | | | | | | | | |
|---|---|---|---|---|---|---|---|---|---|
| | **TVAE** | | **CTGAN** | | **TabDDPM** | | **DGD** | | **Oracle** |
| Metric | Raw | Refined | Raw | Refined | Raw | Refined | Raw | Refined | Real |
| Coverage | $\mathbf{0.9490} \pm 0.0026$ | $0.9382 \pm 0.0031$ | $\mathbf{0.9287} \pm 0.0021$ | $0.8695 \pm 0.0016$ | $\mathbf{0.9498} \pm 0.0013$ | $0.9030 \pm 0.0022$ | $0.9858 \pm 0.0009$ | $\mathbf{0.9697} \pm 0.0092$ | $0.9536 \pm 0.0014$ |
| Avg. Set Size | $10.6940 \pm 0.0391$ | $\mathbf{7.1003} \pm 0.0728$ | $10.5236 \pm 0.0698$ | $\mathbf{5.3499} \pm 0.0268$ | $8.9492 \pm 0.0819$ | $\mathbf{4.9708} \pm 0.0227$ | $14.4019 \pm 0.0534$ | $\mathbf{8.0976} \pm 0.9079$ | $5.0465 \pm 0.0191$ |

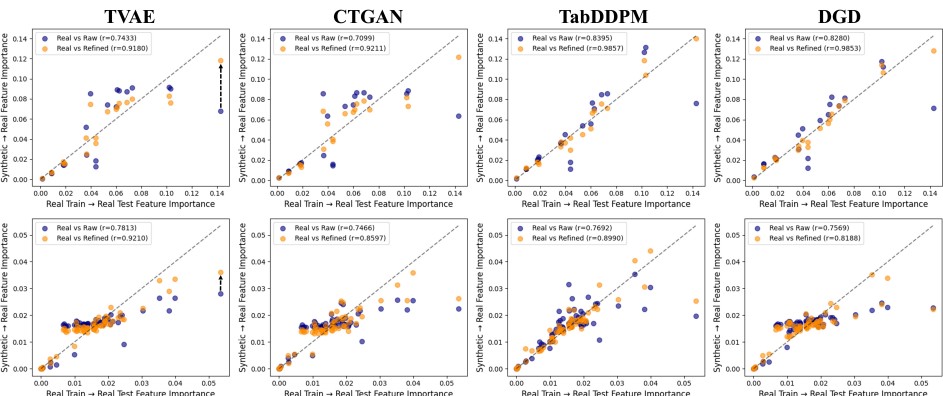

Figure 5: Feature importance correlation analysis. The figure shows the correlation between feature importance rankings derived from models trained on real data versus those trained on synthetic data. Higher correlations indicate better preservation of model interpretability. The first row presents results on TCGA-metadata, and the second row presents results on GOSSIS-1-eICU-cardiovascular.

The feature importance correlation metric measures how well synthetic data preserves the interpretability aspects crucial for healthcare applications. Figure 5 shows that all models demonstrate substantial improvements, with feature importance correlations improving from the 0.71-0.84 range to 0.92-0.99 range for TCGA-metadata, while for GOSSIS-eICU-Cardiovascular dataset, correlations advance from 0.75-0.78 to 0.82-0.92. This indicates that MAPS refined data not only performs better quantitatively but also maintains the feature relationships that clinicians rely on for model interpretation. For instance, examining the feature with the highest importance rank—`tumor_stage` for TCGA-metadata and `d1_arterial_po2_diff` for GOSSIS-eICU-cardiovascular—we observe significant improvements across all models. The arrows demonstrate the progression from raw to refined synthetic data: for TCGA-metadata, the feature importance increases notably from 0.07 to 0.12 after MAPS refinement, approaching the real data value of 0.14. Similarly, for the GOSSIS-eICU-cardiovascular dataset, the importance value is elevated from 0.03 to 0.04, moving closer to the real data's 0.05.

## 4.4 PRIVACY PROTECTION ANALYSIS

Table 4 demonstrates that MAPS maintains decent privacy protections while improving data quality. First, the 0-identifiability filtering in Stage 1 significantly reduces re-identifiability risk in the refined datasets, reducing the identifiability score from an average of 33.62% in raw synthetic datasets (where over one-third of real samples were partnered most closely with synthetic samples) to exactly 0% across all models and datasets. However, membership inference attack (MIA)(El Emam et al., 2022) resistance tests reveal a more nuanced picture. While most results show maintained or improved privacy protection, some exceptions warrant discussion: TabDDPM and DGD show increased MIA recall on TCGA-metadata (from 0.1056 to 0.2159 and 0.0781 to 0.1679 respectively), though reasonable precision values indicate more false positives rather than systematic vulnerabilities. The DOMIAS attack(Van Breugel et al., 2023) results show stable performance near 0.5 (random guessing) across all combinations, indicating MAPS does not introduce density-based privacy vulnerabilities. The overall pattern suggests MAPS's 0-identifiability guarantee provides robust privacy protection, with observed MIA variations likely reflecting the inherent privacy-fidelity tradeoff. For use cases particularly sensitive to membership inference, practitioners may need additional adjustments beyond MAPS, though the modular framework allows tailoring filtering thresholds and evaluation metrics to specific project objectives.

Table 4: Privacy assessment via multiple evaluation metrics. For privacy metrics (Distance-based, MIAs), lower values indicate better privacy protection. IS denotes Identifiability Score measuring re-identification risk. Standard MIA tests basic membership inference, while DOMIAS targets density-based vulnerabilities.

**TCGA-metadata**

| Metric Type | Metric | TVAE | | CTGAN | | TabDDPM | | DGD | |
|---|---|---|---|---|---|---|---|---|---|
| | | Raw | Refined | Raw | Refined | Raw | Refined | Raw | Refined |
| Distance-based | IS | $0.4033\pm_{0.0036}$ | $\mathbf{0.0000}\pm_{0.0000}$ | $0.4331\pm_{0.0027}$ | $\mathbf{0.0000}\pm_{0.0000}$ | $0.4465\pm_{0.0026}$ | $\mathbf{0.0000}\pm_{0.0000}$ | $0.3595\pm_{0.0040}$ | $\mathbf{0.0000}\pm_{0.0000}$ |
| Standard MIA | F1 | $\mathbf{0.3352}\pm_{0.0005}$ | $0.3370\pm_{0.0004}$ | $0.3368\pm_{0.0004}$ | $0.3370\pm_{0.0002}$ | $\mathbf{0.4084}\pm_{0.0012}$ | $0.4555\pm_{0.0036}$ | $\mathbf{0.3906}\pm_{0.0019}$ | $0.4391\pm_{0.0042}$ |
| | Precision | $0.4758\pm_{0.0513}$ | $0.4908\pm_{0.0246}$ | $0.6196\pm_{0.0491}$ | $\mathbf{0.5904}\pm_{0.0472}$ | $0.5046\pm_{0.0048}$ | $0.4983\pm_{0.0072}$ | $0.4887\pm_{0.0121}$ | $0.5048\pm_{0.0106}$ |
| | Recall | $\mathbf{0.0022}\pm_{0.0005}$ | $0.0042\pm_{0.0005}$ | $0.0036\pm_{0.0004}$ | $0.0038\pm_{0.0002}$ | $\mathbf{0.1056}\pm_{0.0026}$ | $0.2159\pm_{0.0053}$ | $\mathbf{0.0781}\pm_{0.0015}$ | $0.1679\pm_{0.0046}$ |
| DOMIAS | Accuracy | $\mathbf{0.4945}\pm_{0.0011}$ | $0.4981\pm_{0.0007}$ | $0.4973\pm_{0.0012}$ | $0.4964\pm_{0.0011}$ | $0.4997\pm_{0.0012}$ | $\mathbf{0.4976}\pm_{0.0010}$ | $\mathbf{0.4975}\pm_{0.0007}$ | $0.5004\pm_{0.0008}$ |
| | AUCROC | $\mathbf{0.4934}\pm_{0.0022}$ | $0.4983\pm_{0.0016}$ | $0.4978\pm_{0.0014}$ | $\mathbf{0.4946}\pm_{0.0011}$ | $0.4989\pm_{0.0009}$ | $0.4975\pm_{0.0004}$ | $\mathbf{0.4946}\pm_{0.0023}$ | $0.4987\pm_{0.0014}$ |

**GOSSIS-1-eICU-cardiovascular**

| Metric Type | Metric | TVAE | | CTGAN | | TabDDPM | | DGD | |
|---|---|---|---|---|---|---|---|---|---|
| | | Raw | Refined | Raw | Refined | Raw | Refined | Raw | Refined |
| Distance-based | IS | $0.4007\pm_{0.0018}$ | $\mathbf{0.0000}\pm_{0.0000}$ | $0.2038\pm_{0.0011}$ | $\mathbf{0.0000}\pm_{0.0000}$ | $0.3429\pm_{0.0037}$ | $\mathbf{0.0000}\pm_{0.0000}$ | $0.0994\pm_{0.0013}$ | $\mathbf{0.0000}\pm_{0.0000}$ |
| Standard MIA | F1 | $0.3334\pm_{0.0001}$ | $0.3335\pm_{0.0001}$ | $0.3343\pm_{0.0002}$ | $\mathbf{0.3340}\pm_{0.0002}$ | $\mathbf{0.3519}\pm_{0.0005}$ | $0.3974\pm_{0.0013}$ | $0.3369\pm_{0.0005}$ | $0.3369\pm_{0.0005}$ |
| | Precision | $0.2000\pm_{0.1458}$ | $0.4283\pm_{0.0811}$ | $0.5797\pm_{0.0716}$ | $0.5318\pm_{0.0570}$ | $0.4902\pm_{0.0102}$ | $0.5147\pm_{0.0050}$ | $0.5560\pm_{0.0473}$ | $0.5346\pm_{0.0273}$ |
| | Recall | $0.0001\pm_{0.0000}$ | $0.0002\pm_{0.0001}$ | $0.0010\pm_{0.0002}$ | $0.0008\pm_{0.0001}$ | $0.0223\pm_{0.0004}$ | $0.0849\pm_{0.0016}$ | $0.0038\pm_{0.0004}$ | $0.0040\pm_{0.0005}$ |
| DOMIAS | Accuracy | $0.4990\pm_{0.0009}$ | $0.4996\pm_{0.0005}$ | $0.4999\pm_{0.0007}$ | $0.4999\pm_{0.0003}$ | $0.4989\pm_{0.0003}$ | $0.4995\pm_{0.0004}$ | $0.4991\pm_{0.0005}$ | $0.4994\pm_{0.0006}$ |
| | AUCROC | $0.4984\pm_{0.0005}$ | $0.4990\pm_{0.0003}$ | $0.4994\pm_{0.0004}$ | $0.4992\pm_{0.0002}$ | $\mathbf{0.4971}\pm_{0.0001}$ | $0.4991\pm_{0.0002}$ | $0.5008\pm_{0.0001}$ | $\mathbf{0.5002}\pm_{0.0003}$ |

## 5 CONCLUSION

We present MAPS, a model-agnostic framework for post-hoc synthetic data refinement that provides formal 0-identifiability guarantees while improving data quality through importance weighting and resampling, with comprehensive evaluation demonstrating consistent improvements in fidelity, utility, and privacy protection across healthcare datasets and multiple generative models.

MAPS's modular design combines sample-level and set-level refinement steps in a unique hierarchical approach, enabling practitioners to customize components according to their requirements. For instance, users can substitute alternative distance measures in Stage 1 to distance to closest record (DCR) (Zhao et al., 2021) or nearest neighbor distance ratio (NNDR) (Zhao et al., 2021) and different classification architectures in Stage 2, and adjust privacy controls from conservative 0-identifiability to alternative $\epsilon$-identifiability levels. This flexibility addresses diverse needs across domains with varying privacy requirements and quality standards. However, it is important to note that while high-quality synthetic data offers greater utility for many applications, it also requires greater responsibility in deployment regardless of privacy guarantees, and a risk assessment should always be conducted that considers the targeted use case for the synthetic dataset (Bartell et al., 2024; Qian et al., 2024; Schmidt et al., 2024). Future work could explore broader applicability to privacy-first generative models such as DP-GAN, PATE-GAN, and ADS-GAN, and extend the framework to handle time-series synthetic data.

ETHICS STATEMENT

This research adheres to the ICLR Code of Ethics. Our work focuses on improving synthetic data quality while strengthening privacy protections, contributing positively to privacy-preserving machine learning. We use publicly available healthcare datasets (TCGA-metadata and GOSSIS-1-eICU-cardiovascular) that have been previously released for research purposes with appropriate ethical approvals. Our MAPS framework enforces 0-identifiability guarantees to minimize re-identification risks, and we conduct comprehensive privacy evaluations including membership inference attack resistance. We acknowledge that even when synthetic data are both high-quality and privacy-preserving, their deployment still requires responsibility. We emphasize throughout the paper that practitioners should conduct risk assessments considering their specific use cases. The synthetic data generation techniques presented should only be applied in contexts where appropriate ethical oversight and data governance frameworks are in place. We have made our implementation publicly available to promote transparency and enable further research in privacy-preserving synthetic data generation.

REPRODUCIBILITY STATEMENT

We have made great efforts to ensure the reproducibility of our results. Complete implementation details are provided in Appendix B, including hyperparameter settings, model configurations, and training procedures for all generative models and the MAPS framework. The full algorithm is detailed in Appendix A, and comprehensive evaluation methodologies are described in Appendix C. Data preprocessing steps are specified in Appendix B.1. The source code for our MAPS framework, experimental setup, and evaluation scripts is available at the anonymous repository `https://anonymous.4open.science/r/MAPS-EBF8`. All experiments use publicly available datasets with clearly documented preprocessing procedures. Statistical significance testing methods and experimental protocols are detailed in the appendix. The modular design of MAPS allows researchers to easily adapt individual components and reproduce results across different generative models and datasets.

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

# Appendix

## A MAPS ALGORITHM

---

**Algorithm 1** MAPS: Model Agnostic Post-hoc Synthetic Data Refinement

---

**Require:** Real dataset $\mathcal{D} = \{x_i\}_{i=1}^N$, Synthetic dataset $\hat{\mathcal{D}} = \{\hat{x}_j\}_{j=1}^M$, Target size $N$
**Ensure:** Refined synthetic dataset $\tilde{\mathcal{D}}$ with $|\tilde{\mathcal{D}}| = N$

 1: **// Stage 1: 0-Identifiability Guarantee**
 2: Compute feature weights $\mathbf{w}$ based on data characteristics
 3: **for** each real sample $x_i \in \mathcal{D}$ **do**
 4:    $r_i \leftarrow \min_{x_j \in \mathcal{D} \setminus \{x_i\}} \|\mathbf{w} \cdot (x_i - x_j)\|$ {Distinctness threshold}
 5: **end for**
 6: $\mathcal{D}_{filtered} \leftarrow \emptyset$ {Initialize filtered synthetic dataset}
 7: **for** each synthetic sample $\hat{x}_j \in \hat{\mathcal{D}}$ **do**
 8:    $is\_safe \leftarrow True$
 9:    **for** each real sample $x_i \in \mathcal{D}$ **do**
10:      **if** $\|\mathbf{w} \cdot (x_i - \hat{x}_j)\| < r_i$ **then**
11:        $is\_safe \leftarrow False$ {Too close to real sample}
12:        **break**
13:      **end if**
14:    **end for**
15:    **if** $is\_safe$ **then**
16:      $\mathcal{D}_{filtered} \leftarrow \mathcal{D}_{filtered} \cup \{\hat{x}_j\}$
17:    **end if**
18: **end for**
19: **// Stage 2: Fidelity Enhancement via Importance Weighting**
20: Randomly sample $\mathcal{D}_{filtered\_train} \subset \mathcal{D}_{filtered}$ with $|\mathcal{D}_{filtered\_train}| = N$
21: $\mathcal{D}_{remaining} \leftarrow \mathcal{D}_{filtered} \setminus \mathcal{D}_{filtered\_train}$ {Samples for SIR}
22: Create training set: $\mathcal{X}_{train} = \mathcal{D} \cup \mathcal{D}_{filtered\_train}$
23: Create labels: $y_i = 1$ for $x_i \in \mathcal{D}$, $y_j = 0$ for $\hat{x}_j \in \mathcal{D}_{filtered\_train}$
24: Train binary classifier $c_\phi$ on $(\mathcal{X}_{train}, \mathbf{y})$
25: $\pi_1 \leftarrow |\mathcal{D}|/|\mathcal{X}_{train}|, \pi_0 \leftarrow |\mathcal{D}_{filtered\_train}|/|\mathcal{X}_{train}|$
26: **for** each $\hat{x}_j \in \mathcal{D}_{remaining}$ **do**
27:    $p_j \leftarrow c_\phi(\hat{x}_j)$ {Probability of being real}
28:    $\hat{w}_j \leftarrow \frac{\pi_0}{\pi_1} \cdot \frac{p_j}{1-p_j}$ {Importance weight}
29: **end for**
30: **// Stage 3: Sampling-Importance-Resampling (SIR)**
31: $W_{total} \leftarrow \sum_{\hat{x}_j \in \mathcal{D}_{remaining}} \hat{w}_j$
32: **for** each $\hat{x}_j \in \mathcal{D}_{remaining}$ **do**
33:    $p_j^{norm} \leftarrow \hat{w}_j / W_{total}$ {Normalized sampling probability}
34: **end for**
35: $\tilde{\mathcal{D}} \leftarrow \emptyset$
36: **for** $n = 1$ to $N$ **do**
37:    Sample index $j$ from $\mathcal{D}_{remaining}$ with probabilities $\{p_j^{norm}\}$
38:    $\tilde{\mathcal{D}} \leftarrow \tilde{\mathcal{D}} \cup \{\hat{x}_j\}$
39: **end for**
40: **return** $\tilde{\mathcal{D}}$

---

## B IMPLEMENTATION DETAILS

### B.1 DATA PREPROCESSING

For TCGA-metadata, we selected 21 variables with missing value percentages below 50%, and remaining missing values are subsequently imputed using the ICE imputer from the HyperImpute

library (Jarrett et al., 2022) before generative model training. For GOSSIS-1-eICU-cardiovascular, we utilize the preprocessed `gossis-1-eicu-only-model-ready.csv.gz` file (Raffa et al., 2022), which contains no missing values.

## B.2 MODEL TRAINING

All generative models except DGD are implemented using the Synthcity library (Qian et al., 2023) with default hyperparameters. TabDDPM is configured with 10,000 training iterations, while DGD is trained with a learning rate of 1e-2, 50 latent dimensions, and 50 components. For TCGA-metadata, DGD uses 2000 epochs; for GOSSIS-1-eICU-cardiovascular, this is reduced to 1000 epochs. All experiments employ a 60:40 train-test split.

## B.3 FIDELITY CLASSIFIER TRAINING

The fidelity classifier employs a 2-layer MLP architecture with hidden layer sizes adapted to dataset characteristics: 240×120 for TCGA-metadata and 480×240 for GOSSIS-1-eICU-cardiovascular. During training, we use balanced sampling (N real samples, N synthetic samples) to ensure $\pi_0/\pi_1 = 1$ in Equation (5). The training data is split 80:20 for training and evaluation. For data preprocessing prior to training, the classifier uses min-max normalization for numerical variables and integer encoding for categorical variables. Training employs the Adam optimizer with learning rate 0.001, batch size 64, and early stopping with patience of 10 epochs to prevent overfitting.

## B.4 IMPORTANCE WEIGHTS POST-PROCESSING

The output of the fidelity classifier provides our importance weights, which are subsequently normalized and used for probability-proportional sampling. However, raw importance weights often exhibit significant skewness, where the majority of synthetic samples receive very small weights while a few samples obtain disproportionately large weights. This distribution can lead to problematic oversampling of only a handful of synthetic samples, thereby failing to capture the holistic distributional properties of the real data and potentially increasing sampling variance.

To address this issue, we employ a flattening transformation as suggested by (Grover et al., 2019). The flattening process applies a power transformation to the raw importance weights:

$$\hat{w}_{\text{flattened}}(x) = \hat{w}_\phi(x)^\alpha \qquad (8)$$

where $\hat{w}_\phi(x)$ represents the original importance weight estimated by the fidelity classifier, and $\alpha$ is a flattening parameter that controls the degree of variance reduction. When $\alpha = 1$, the weights remain unchanged, while smaller values of $\alpha$ increasingly flatten the weight distribution by compressing the range between high and low weights, conversely, larger values of $\alpha$ amplify the differences between weights.

This transformation serves multiple purposes: (1) it tunes the variance of importance weights, leading to more controllable sampling behavior; (2) it prevents extreme weights from dominating the resampling process; and (3) it ensures broader coverage of the synthetic data space while maintaining the relative preference for higher-fidelity samples. The flattening parameter $\alpha$ provides a tunable trade-off between importance weighting effectiveness and sampling diversity.

In our implementation, we employ dataset-specific flattening parameters determined empirically: $\alpha = 1.4$ for all generative models on the TCGA-metadata dataset, and $\alpha = 0.8$ for all models on the GOSSIS-1-eICU-cardiovascular dataset.

## C EVALUATION DETAILS

To comprehensively assess the effectiveness of MAPS across multiple dimensions of synthetic data quality, we employ a multi-faceted evaluation framework that systematically measures improvements in distributional fidelity, downstream task utility, and privacy protection. Our evaluation protocol is designed to capture both marginal and joint distributional properties while ensuring that improvements translate to practical utility in real-world applications. The evaluation framework encompasses three

primary dimensions: (1) fidelity metrics that quantify how well synthetic data approximates the statistical properties of real data, (2) utility metrics that assess performance on downstream tasks using a rigorous "train-on-synthetic, test-on-real" protocol, and (3) privacy metrics that ensure refinement does not compromise data privacy protections.

## C.1 FIDELITY EVALUATION METRICS

Fidelity evaluation forms the cornerstone of our assessment framework, as it directly measures whether MAPS successfully improves the statistical alignment between synthetic and real data. We employ a comprehensive suite of distributional similarity metrics that capture both marginal and joint distributional properties across mixed-type tabular data. The selection of these metrics follows established practices in synthetic data evaluation literature (Qian et al., 2023; Lautrup et al., 2024; Kotelnikov et al., 2023), ensuring comprehensive coverage of distributional aspects critical for tabular synthetic data quality assessment. This multi-metric approach ensures robust evaluation across different aspects of distributional fidelity, from univariate marginals to complex multivariate relationships.

### C.1.1 QUANTIFYING THE MARGINAL DISTRIBUTION SIMILARITY

We employ 6 complementary metrics to assess marginal distribution similarity, each targeting specific aspects of distributional alignment:

**Kolmogorov-Smirnov Test Statistic:** This non-parametric test measures the maximum absolute difference between cumulative distribution functions of real and synthetic data. Applied exclusively to numerical variables, it ranges from 0 to 1, where higher values indicate better distributional alignment (we report 1 - KS statistic for interpretability).

**Chi-square Test:** This statistical test evaluates the independence hypothesis between real and synthetic categorical distributions through contingency table analysis. Applied to categorical variables, p-values range from 0 to 1, where higher p-values mean weaker evidence against the null hypothesis of distributional equality. When p-values are high, we fail to reject the null hypothesis, suggesting insufficient evidence to conclude that the distributions differ significantly.

**Jensen-Shannon Distance:** This symmetric divergence measure quantifies the similarity between two probability distributions as the square root of Jensen-Shannon divergence. Applied to categorical variables, it ranges from 0 to 1, where lower values indicate better similarity.

**Total Variation Distance:** This metric computes half of the L1 distance between two probability mass functions, representing the overall difference in probability assignments. Applied to categorical variables, it ranges from 0 to 1, where lower values indicate better similarity.

**Hellinger Distance:** This metric measures the similarity between probability distributions based on the Euclidean distance between their square-rooted probability vectors. Applied to categorical variables, it ranges from 0 to 1, where lower values indicate better similarity.

**Inverse KL Divergence:** This metric transforms the Kullback-Leibler divergence using the formula $1/(1 + \text{KL}(P||Q))$ to provide a bounded similarity measure. Applied to categorical variables, it ranges from 0 to 1, where higher values indicate better similarity.

These 6 metrics collectively provide a comprehensive assessment of marginal distribution similarity by capturing different aspects of distributional alignment across both numerical and categorical variables, ensuring robust evaluation of synthetic data fidelity at the univariate level.

### C.1.2 QUANTIFYING THE JOINT DISTRIBUTION SIMILARITY

For joint distribution assessment, we employ two specialized metrics that capture multivariate relationships:

**Wasserstein Distance (WD):** This optimal transport-based metric measures the minimum cost of transforming one distribution into another in the joint numerical feature space. Applied to all numerical variables simultaneously using the Sinkhorn algorithm for computational efficiency, it ranges from 0 to infinity, where lower values indicate better similarity. The implementation is

provided by the GeomLoss package (Feydy et al., 2019) for efficient computation of optimal transport distances.

**Joint Jensen-Shannon Distance (JSD):** This metric extends Jensen-Shannon divergence to joint categorical distributions by computing divergence over the Cartesian product of all categorical variable combinations. Applied to all categorical variables simultaneously, it ranges from 0 to 1, where lower values indicate better similarity.

These two joint distribution metrics complement the marginal assessments by capturing multivariate dependencies and interaction patterns that are crucial for downstream task performance, providing a complete picture of distributional fidelity across the feature space.

## C.2 MIXED-TYPE CORRELATION MATRIX

We compute correlation matrices using Spearman correlation coefficients for numerical-numerical relationships, Cramér's V for categorical-categorical associations, and correlation ratios for categorical-numerical relationships. The association matrix is not symmetrical due to the asymmetry of correlation ratios.

**Normalized Frobenius Norm (NFN):** This metric quantifies the difference between correlation structures by computing the Frobenius norm of the difference between real and synthetic correlation matrices, normalized by the square root of the total number of matrix elements. It ranges from 0 to infinity, where lower values indicate better correlation structure preservation.

## C.3 UTILITY EVALUATION PROTOCOL

Our utility evaluation employs a rigorous "train-on-synthetic, test-on-real" protocol that directly assesses whether synthetic data can replace real data for downstream applications. This evaluation strategy reflects real-world usage scenarios where synthetic data would be used for model development before deployment on real data.

For clustering evaluation, we use K-means with k=5 clusters (determined by elbow analysis) on numerical variables only. The evaluation protocol splits real data 80:20, with clustering models trained on synthetic data and evaluated on the real test set using Adjusted Rand Index (ARI) and Adjusted Mutual Information (AMI) metrics against true class labels. These metrics range from 0 to 1 (with ARI potentially negative for very poor clusterings), where higher values indicate better clustering agreement with ground truth labels.

Classification tasks employ Random Forest with 100 estimators, using the same train-test split strategy. For TCGA-metadata, the target variable is `cancer_types` while for GOSSIS-1-eICU-cardiovascular, the target variable is `dx_class`. We evaluate performance using accuracy, F1-score (macro-averaged), and ROC-AUC metrics, all ranging from 0 to 1 with higher values indicating better performance.

Building on the classification tasks, we evaluate uncertainty quantification capabilities through split conformal prediction (Angelopoulos et al., 2024), a variant of conformal prediction that uses a separate calibration set, following a "train-on-synthetic, calibrate-on-synthetic, test-on-real" protocol. We train classifiers on synthetic training data, then use a synthetic calibration set to compute non-conformity scores using the inverse probability score function:

$$s(x, y) = 1 - f_\phi(x)[y] \tag{9}$$

where $f_\phi(x)[y]$ is the predicted probability for true class $y$. For miscoverage level $\alpha = 0.05$ (target coverage 95%), we construct prediction sets:

$$C(x_{test}) = \{c : 1 - f_\phi(x_{test})[c] \le \hat{q}\} \tag{10}$$

where $\hat{q}$ is the empirical $(1 - \alpha)$-quantile of calibration scores. We assess performance using coverage (fraction of test samples where true labels are in prediction sets) and average set size (efficiency measure), comparing against oracle performance where both training and calibration use

real data. This evaluation directly addresses whether synthetic data can provide reliable uncertainty quantification for real-world deployment, which is critical for high-stakes applications like healthcare.

Feature importance correlation measures Spearman correlation between importance rankings derived from models trained on real data versus synthetic data, providing insight into interpretability preservation. This metric ranges from -1 to 1, where values closer to 1 indicate better preservation of feature relationships crucial for model interpretation. We included this metric because maintaining interpretable feature relationships is critical for applications in sensitive domains like healthcare, where practitioners require consistent and explainable model behavior.

### C.4 Privacy Evaluation Protocol

Privacy assessment employs multiple approaches to ensure comprehensive evaluation of data protection guarantees. We evaluate both formal identifiability guarantees through identifiability measures and empirical privacy resistance through attack-based assessments.

**Identifiability Score (IS):** This metric directly measures the proportion of real samples that can be identified through synthetic data, computed as the fraction of real samples whose nearest synthetic neighbor is closer than their nearest real neighbor. Values range from 0 to 1, where 0 indicates perfect 0-identifiability (no real samples pair with a synthetic sample more closely than they pair with a real sample), a value of 1 indicates maximum identifiability risk, where every real sample has a synthetic counterpart that is closer to it than its nearest real neighbor. The IS implementation is provided by the Synthcity library (Qian et al., 2023).

**Membership Inference Attack (MIA) Resistance:** We evaluate resistance to standard membership inference attacks using precision, recall, and F1-score metrics, where lower values indicate better privacy protection. These attacks attempt to determine whether specific samples were included in the training data used to generate synthetic samples. The standard MIA implementation is provided by the SynthEval package (Lautrup et al., 2024), with evaluation conducted using stratified cross-validation with 5 folds to ensure robust privacy assessment across different data splits.

**DOMIAS Attack Resistance:** We assess resistance to Density-based Membership Inference Attacks through accuracy and AUC-ROC metrics, where values closer to 0.5 (random guessing) indicate better privacy protection. This specialized attack targets density-based vulnerabilities. The DOMIAS attack implementation is provided by the Synthcity package (Qian et al., 2023).

## D  PCA plots

The PCA visualizations in Figures 6 and 7 demonstrate how MAPS refinement brings the synthetic data distribution closer to the real data distribution in the principal component space, providing visual confirmation of the quantitative improvements observed in our fidelity metrics.

## E  Statistical Significance Analysis

Statistical significance testing was performed using paired t-tests across 5 experimental runs. The paired t-test is appropriate for our experimental design as it compares performance between raw and refined synthetic data on the same underlying generative models and datasets.

For each metric, we computed the mean improvement, standard deviation, and p-values across independent experimental runs. Results with $p < 0.05$ are considered statistically significant.

## F  Detailed Utility Results

Table 5 shows the detailed numerical utility results corresponding to Figure 4.

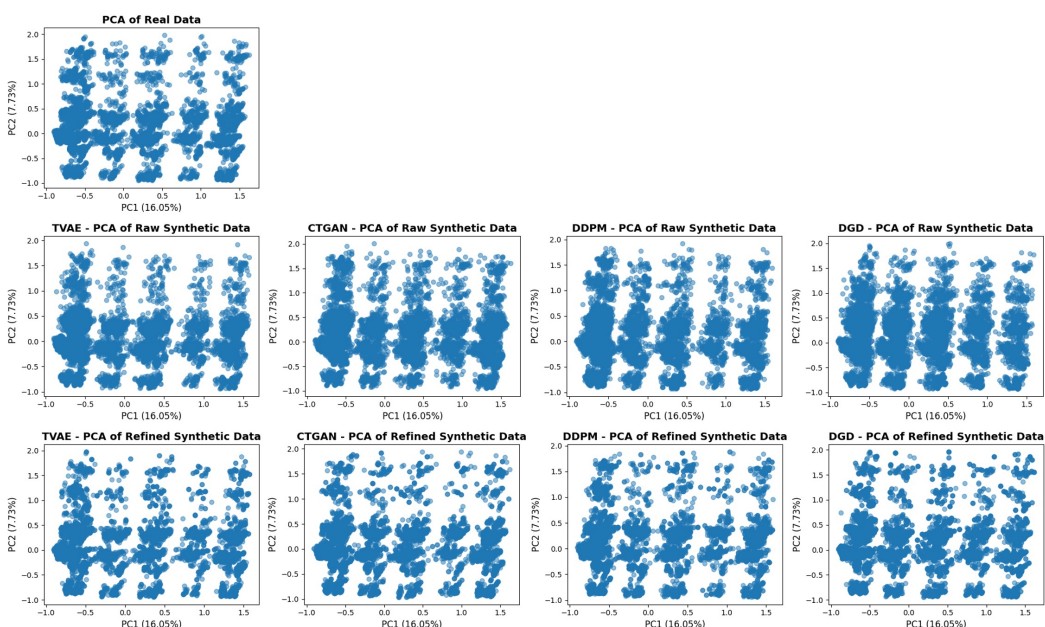

Figure 6: PCA visualization of TCGA-metadata dataset showing the distribution of real data, raw synthetic data, and refined synthetic data in the first two principal components.

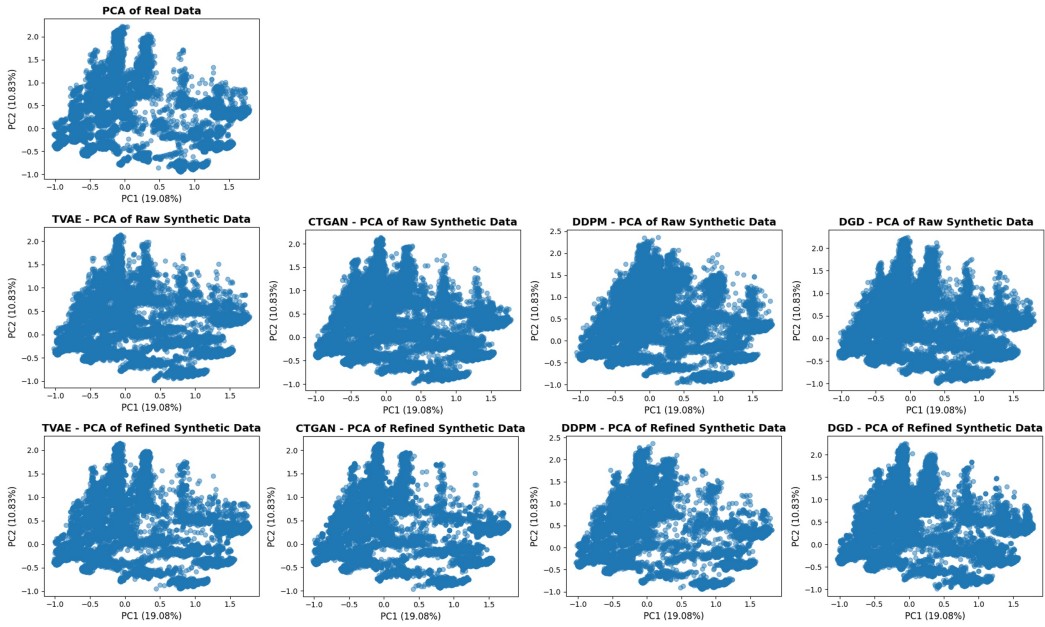

Figure 7: PCA visualization of GOSSIS-1-eICU-cardiovascular dataset showing the distribution of real data, raw synthetic data, and refined synthetic data in the first two principal components.

Table 5: Comprehensive downstream task performance analysis. Results demonstrate MAPS effectiveness across clustering, classification, and feature importance preservation tasks using the "train-on-synthetic, test-on-real" evaluation protocol. The Oracle column represents the upper-bound performance achieved by training and testing on real data, serving as the ground truth benchmark for these tasks. * Indicates insufficient data for significance testing.

**TCGA-metadata**

| Task | Metric | TVAE | | CTGAN | | TabDDPM | | DGD | | Oracle |
|---|---|---|---|---|---|---|---|---|---|---|
| | | Raw | Refined | Raw | Refined | Raw | Refined | Raw | Refined | Real |
| **K-Means** | **ARI** | 0.7458±0.1759 | **0.8921**±0.0082 | 0.6069±0.0163 | **0.8706**±0.0107 | 0.0022±0.0006 | **0.7384**±0.0127 | **0.7965**±0.0052 | 0.6950±0.0181 | 1.0000 |
| | **AMI** | 0.7654±0.1011 | **0.8740**±0.0057 | 0.5815±0.0071 | **0.8543**±0.0070 | 0.0020±0.0004 | **0.6923**±0.0129 | **0.7224**±0.0075 | 0.6465±0.0222 | 1.0000 |
| **Classification** | **Accuracy** | 0.2780±0.0160 | **0.5398**±0.0157 | 0.2192±0.0058 | **0.4807**±0.0083 | 0.1742±0.0136 | **0.4775**±0.0074 | 0.3749±0.0040 | **0.5173**±0.0167 | 0.6589±0.0079 |
| | **F1 Score** | 0.1647±0.0101 | **0.3848**±0.0234 | 0.1152±0.0044 | **0.3043**±0.0049 | 0.0866±0.0145 | **0.3372**±0.0195 | 0.2400±0.0093 | **0.3707**±0.0189 | 0.5576±0.0081 |
| | **ROC-AUC** | 0.7664±0.0010* | **0.8914**±0.0077 | 0.7022±0.0034 | **0.8547**±0.0080 | 0.6272±0.0114 | **0.8804**±0.0035 | 0.8234±0.0083 | **0.9082**±0.0037 | 0.9590±0.0027 |
| | **Feat. Imp. Corr.** | 0.7393±0.0035 | **0.9168**±0.0037 | 0.7067±0.0062 | **0.9199**±0.0053 | 0.8429±0.0030 | **0.9854**±0.0012 | 0.8268±0.0037 | **0.9850**±0.0017 | 1.0000 |

**GOSSIS-1-eICU-cardiovascular**

| Task | Metric | TVAE | | CTGAN | | TabDDPM | | DGD | | Oracle |
|---|---|---|---|---|---|---|---|---|---|---|
| | | Raw | Refined | Raw | Refined | Raw | Refined | Raw | Refined | Real |
| **K-Means** | **ARI** | 0.5257±0.0572 | **0.5810**±0.0977 | 0.4345±0.0015 | **0.5476**±0.0192 | 0.0000±0.0000 | **0.3075**±0.0043 | 0.2569±0.0079 | **0.4287**±0.0055 | 1.0000 |
| | **AMI** | 0.5380±0.0261 | **0.5923**±0.0692 | 0.5272±0.0031 | **0.5848**±0.0156 | 0.0000±0.0000 | **0.3879**±0.0038 | 0.3236±0.0068 | **0.5472**±0.0116 | 1.0000 |
| **Classification** | **Accuracy** | 0.3745±0.0063 | **0.5008**±0.0031 | 0.3763±0.0042 | **0.4545**±0.0035 | 0.3574±0.0073 | **0.4726**±0.0039 | 0.4476±0.0086 | 0.4593±0.0107 | 0.5922±0.0023 |
| | **F1 Score** | 0.1867±0.0043 | **0.2497**±0.0016 | 0.1812±0.0067 | **0.2144**±0.0019 | 0.1287±0.0043 | **0.2104**±0.0022 | 0.2085±0.0080 | **0.2466**±0.0041 | 0.3234±0.0020 |
| | **ROC-AUC** | 0.7895±0.0058 | **0.8329**±0.0022 | 0.7450±0.0027* | 0.7867±0.0033 | 0.7722±0.0042 | **0.7837**±0.0057 | 0.8308±0.0029 | **0.8711**±0.0051 | 0.9099±0.0021 |
| | **Feat. Imp. Corr.** | 0.7821±0.0020 | **0.9197**±0.0018 | 0.7492±0.0045 | **0.8511**±0.0058 | 0.7597±0.0087 | **0.8863**±0.0074 | 0.7497±0.0049 | **0.8239**±0.0043 | 1.0000 |

# G MORTALITY PREDICTION

To complement the multi-class prediction results presented in the main paper, we evaluate MAPS on mortality prediction, a critical task in clinical research and patient care. While the main results demonstrate MAPS effectiveness on the more challenging multi-class problem, these mortality prediction results confirm that the framework also provides consistent improvements on this important binary classification task, further validating its broad utility enhancement across different tasks. See below Table 6 for detailed results.

Table 6: Mortality prediction performance on **TCGA-metadata** and **GOSSIS-1-eICU-cardiovascular**. The Oracle column represents the upper-bound performance achieved by training and testing on real data, serving as the ground truth benchmark for these tasks.

**TCGA-metadata**

| Task | Metric | TVAE | | CTGAN | | TabDDPM | | DGD | | Oracle |
|---|---|---|---|---|---|---|---|---|---|---|
| | | Raw | Refined | Raw | Refined | Raw | Refined | Raw | Refined | Real |
| **Mortality Prediction** | **Accuracy** | 0.8293±0.0061 | 0.8307±0.0046 | 0.8245±0.0023 | 0.8308±0.0062 | 0.8346±0.0068 | 0.8370±0.0041 | 0.8010±0.0039 | **0.8184**±0.0032 | 0.8563±0.0028 |
| | **F1 Score** | 0.7871±0.0099 | 0.7939±0.0060 | 0.7933±0.0031 | 0.7937±0.0088 | 0.8003±0.0094 | 0.8035±0.0057 | 0.7364±0.0095 | **0.7772**±0.0058 | 0.8273±0.0029 |
| | **ROC-AUC** | 0.8847±0.0048 | 0.8853±0.0043 | 0.8716±0.0062 | **0.8790**±0.0081 | 0.8875±0.0075 | 0.8889±0.0044 | 0.8625±0.0075 | **0.8730**±0.0045 | 0.9103±0.0034 |
| | **Feat. Imp. Corr.** | 0.9662±0.0041 | 0.9694±0.0031 | 0.9546±0.0083 | **0.9678**±0.0072 | 0.9822±0.0091 | 0.9885±0.0105 | 0.4319±0.0322 | **0.9617**±0.0056 | 1.0000 |

**GOSSIS-1-eICU-cardiovascular**

| Task | Metric | TVAE | | CTGAN | | TabDDPM | | DGD | | Oracle |
|---|---|---|---|---|---|---|---|---|---|---|
| | | Raw | Refined | Raw | Refined | Raw | Refined | Raw | Refined | Real |
| **Mortality Prediction** | **Accuracy** | 0.9361±0.0017 | **0.9394**±0.0011 | 0.9350±0.0021 | 0.9363±0.0019 | 0.9366±0.0007 | **0.9393**±0.0010 | 0.9285±0.0017 | **0.9371**±0.0010 | 0.9405±0.0010 |
| | **F1 Score** | 0.7191±0.0132 | **0.7550**±0.0067 | 0.7270±0.0144 | **0.7422**±0.0118 | 0.7212±0.0068 | **0.7617**±0.0056 | 0.6207±0.0176 | **0.7236**±0.0049 | 0.7533±0.0073 |
| | **ROC-AUC** | 0.9048±0.0047 | **0.9096**±0.0052 | 0.9016±0.0036 | 0.9027±0.0055 | 0.9150±0.0055 | 0.9167±0.0045 | 0.8965±0.0063 | **0.9112**±0.0053 | 0.9172±0.0038 |
| | **Feat. Imp. Corr.** | 0.9062±0.0175 | **0.9662**±0.0093 | 0.8749±0.0155 | **0.9182**±0.0208 | 0.5222±0.0865 | **0.9662**±0.0145 | 0.8657±0.0158 | **0.9563**±0.0096 | 1.0000 |

# H LARGE LANGUAGE MODEL USAGE STATEMENT

In accordance with ICLR 2026 submission guidelines, we disclose the use of Large Language Models in the preparation of this manuscript. We utilized Claude Sonnet 4 (Anthropic) exclusively for writing assistance and language polishing purposes. Specifically, the model was employed to improve sentence structure, enhance clarity, and correct grammar. All technical content, methodology, experimental design, results, and scientific contributions remain entirely our own work.

