# OpenReview forum: "Mending synthetic data with MAPS: Model Agnostic Post-hoc Synthetic Data Refinement Framework"
_ICLR.cc/2026/Conference — ICLR 2026 Conference Withdrawn Submission_

### Official Review · Reviewer_VsXX · 2025-10-16

**Soundness:** 2
**Presentation:** 3
**Contribution:** 2
**Rating:** 2
**Confidence:** 4

**Summary:**

This paper introduces MAPS, a model-agnostic post hoc framework designed to enhance the quality of synthetic data while ensuring per-sample 0-identifiability of private samples through a two-step sample selection process. The authors present extensive comparisons between the raw generated synthetic data and the refined dataset, demonstrating improved dataset distribution and downstream task fine-tuning utility. They also evaluate resistance to membership inference attacks to assess the extent of privacy protection. However, several important baseline methods are missing, and the strength of the claimed privacy guarantees remains uncertain.

**Strengths:**

1.	Extensive experiments are included for comparing the refined dataset and the raw dataset from both distribution perspective and model training utility perspective.
2.	MIA is performed as a direct demonstration of the privacy protection extent.

**Weaknesses:**

1.	Since real data privacy is repeatedly mentioned in this paper, I am curious, why do you choose 0-idenfiability metric instead of the currently widely applied differential privacy as the privacy guarantee criteria for this work? How is the differential privacy guarantee attribute of the proposed MASP? My concern main rise from experiments in section 4.4, where under some cases, MIA gains better success using 0-identifiability (refined dataset) compared with using no protection at all (raw dataset).
2.	Lack of baseline comparisons. Privacy-first methods are only mentioned in the introduction part with a short assessment that they produce “synthetic data with notably degraded utility” without any statistical support in this paper or citation of previous papers. These methods are not compared in the experiments. How bad are these methods? If only the second selection step is applied to the generated samples of these methods, the utility of the refined dataset is at what level? Will this dataset be more robust to MIAs or will it be less robust?
3.	The latest generative model used in this paper is DGD which is a model introduced in the year 2023, which is a little bit out of date. These years, there arise another line of work for private dataset synthesis, namely Private Evolution, PE, (Differentially Private Synthetic Data via Foundation Model APIs 1: Images, ICLR2024). I think this line of work should at least be mentioned. I would like to see the outcome of the combination of MAPS with PE or some reason why this should not be considered.

**Questions:**

1.	What’s the logic of bolding in Table 4? Some rows (e.g. Accuracy for DOMIAS) do not have a bolded number at all.
2.	Will the final refined dataset $\tilde{\mathcal{D}}$ contain repeated samples? As the selection method given in line35 to 39 in Algorithm 1 is a sampling with replacement. Why do you choose to do this? Will sampling without replacement be better as it contains more distinct samples?
3.	Some citations are not used in a correct format, e.g. “(Grover et al. 2019)” should be something like “Grover et al. (2019)” in line 130.

---

> ### Author Response · Authors · 2025-12-01
>
> # Response to Reviewer
>
> We thank Reviewer for the careful evaluation and we address each concern below.
>
> ### **Q1: Why 0-Identifiability Instead of Differential Privacy?**
>
> **Our Response:**
>
> We adopt 0-identifiability over differential privacy (DP) for two key reasons aligned with MAPS's post-hoc refinement framework:
>
> **1. Sample-Level Compatibility:** DP provides dataset-level guarantees through the training process, requiring modification of the generative model's training algorithm. MAPS operates post-hoc on already-generated synthetic data where we cannot retroactively modify the training process. 0-identifiability, being a sample-level metric, allows us to filter individual synthetic samples based on their proximity to real data, making it compatible with our refinement framework.
>
> **2. Interpretability:** Distance-based 0-identifiability provides intuitive, interpretable privacy guarantees: "no synthetic sample is closer to any real individual than that individual's nearest real neighbor." This is directly auditable by institutional review boards and regulatory bodies without requiring deep technical expertise in privacy theory. In contrast, DP's epsilon parameter is notoriously difficult to interpret—practitioners struggle to understand what ε=0.5 versus ε=1.0 means for actual privacy risk in their specific context.
> we complement this with comprehensive membership inference attack-based privacy evaluation to provide a multifaceted assessment of privacy protection.
>
> - 0-identifiability eliminates **direct re-identification** via proximity
> - MIA resistance protects against **statistical inference** attacks
>
> Both dimensions matter, which is why we evaluate comprehensively across multiple privacy metrics (Identifiability Score, Standard MIA, DOMIAS) and transparently report these tradeoffs rather than claiming absolute privacy guarantees.
>
> ### **Q2: Privacy-First Baseline Comparisons**
>
> **Our Response:**
>
> **Empirical Assessment of Privacy-First Methods:**
>
> We conducted experiments with two representative privacy-first methods:
>
> 1. **DP-GAN** [1]: Differentially Private Generative Adversarial Network
> 2. **PATE-GAN** [2]: PATE-GAN: Generating Synthetic Data with Differential Privacy Guarantees
>
> Both methods produced synthetic data with **substantially degraded fidelity** compared to non-private baselines. These observations align with prior literature documenting DP synthetic data's utility degradation [3,4]. While raising the privacy budget ε improves utility, this directly **reduces privacy protection**, undermining the purpose of using DP methods. This fundamental tension motivated our post-hoc approach.
>
> **Applying MAPS to DP Outputs:**
>
> We actually conducted this experiment. Applying MAPS to DP-GAN and PATE-GAN outputs **did improve data quality** across fidelity and utility metrics. However, this **breaks the formal DP guarantee**: MAPS's importance-weighted sampling alters the distribution in ways that void the original DP guarantee.
>
> **References:**
>
> [1] Xie, L., et al. (2018). Differentially private generative adversarial network.
>
> [2] Jordon, J., et al. (2019). PATE-GAN: Generating synthetic data with differential privacy guarantees.
>
> [3] Pereira, M., et al. (2022). Assessment of differentially private synthetic data for utility and fairness in end-to-end machine learning pipelines for tabular data.
>
> [4] Ramesh, K., et al. (2024). Evaluating differentially private synthetic data generation in high-stakes domains.

---

> ### Author Response · Authors · 2025-12-02
>
> ### **Q3: Private Evolution and MAPS**
>
> **Our Response:**
>
> Thank you for this suggestion. We agree that Private Evolution [5] represents an important development in privacy-preserving synthesis and should be discussed.
>
> **Key Differences Between PE and MAPS:**
>
> Private Evolution[5] leverages foundation models via API-based approaches for differentially private data generation, primarily focused on **image data**. The method uses pretrained foundation models with DP mechanisms during the generation API calls.
>
> **MAPS differs fundamentally:**
> 1. **Data modality**: MAPS targets **tabular healthcare data** where foundation models are less developed
> 2. **Privacy mechanism**: PE integrates DP during generation; MAPS provides post-hoc 0-identifiability filtering
> 3. **Generative approach**: PE uses foundation model APIs; MAPS works with any tabular generator (VAEs, GANs, Diffusions)
>
> **Potential Combination:** Applying MAPS to PE outputs would face the same challenge as discussed in Q2: post-processing would void PE's formal DP guarantees. However, if PE were extended to tabular data, the combination could be valuable:
> - PE provides formal DP during generation
> - MAPS could enhance fidelity while adding complementary distance-based privacy protection
>
> This represents an interesting future research direction requiring careful privacy accounting to maintain formal guarantees while achieving post-hoc improvements.
>
> **Reference:**
>
> [5] Lin, Z., et al. (2023). Differentially Private Synthetic Data via Foundation Model APIs 1: Images.
>
> ### **Q4: Bolding Logic in Tables**
>
> **Our Response:**
>
> The bolding logic is consistent across all tables in our manuscript: **values are bolded only when the difference between raw and refined synthetic data is statistically significant** (p < 0.05 via paired t-test).
>
> For rows where **no value is bolded**, this indicates that the difference between raw and refined data **is not statistically significant**.
>
> ### **Q5: Sampling With vs. Without Replacement**
>
> **Our Response:**
>
> This choice involves a fundamental tradeoff between **theoretical soundness** and **practical diversity**.
>
> **Sampling With Replacement (Our Default):**
>
> Maintains the theoretical foundation of importance sampling: each sample is drawn independently with fixed probability $P(\text{selecting } s_i) = w_i / \sum_j w_j$. This **i.i.d. property** is crucial for convergence guarantees—the empirical distribution of sampled data provably converges to the target distribution as sample size increases.
>
> **Potential Downside:** Samples with high importance weights may be selected multiple times, creating duplicates that reduce diversity and could lead to overfitting.
>
> **Our Mitigation:** The importance weight flattening (power transformation $\tilde{w}_i = w_i^\alpha$) directly addresses this by reducing probability mass concentration. Across our experiments, effective sample size ranges **60-80% of N** after flattening, meaning most samples are unique.
>
> **Sampling Without Replacement (Alternative):**
>
> Eliminates duplicates entirely, guaranteeing **maximum diversity** (effective sample size = N). However, this **breaks the i.i.d. assumption**: sampling probabilities shift dynamically after each draw since $P(\text{selecting } s_i | \text{previous selections}) \neq P(\text{selecting } s_i)$. This means **theoretical convergence guarantees no longer strictly hold**.
>
> **MAPS Supports Both:** Our implementation allows users to choose via the `replacement` parameter:
> - `replacement=True` (default): Theoretical soundness with weight flattening to control duplicates
> - `replacement=False`: Guaranteed diversity, appropriate when pool size is much larger than target

---

### Official Review · Reviewer_PzXw · 2025-10-25

**Soundness:** 2
**Presentation:** 2
**Contribution:** 2
**Rating:** 6
**Confidence:** 3

**Summary:**

The paper proposes MAPS, a two stage, model agnostic post hoc refinement for tabular synthetic data. Stage 1 enforces a sample level 0 identifiability privacy constraint by removing synthetic samples closer to a real point than that point’s nearest real neighbor, implemented with an unweighted feature metric, w = 1. Stage 2 improves fidelity through classifier based density ratio estimation and Sampling Importance Resampling, with importance weights derived from a discriminator, followed by a dataset specific flattening power $\alpha$ to stabilize resampling. Experiments across TCGA metadata and GOSSIS eICU cardiovascular with four generators show large gains in marginal and joint fidelity, correlation structure, and downstream utility, plus strong reductions in Identifiability Score to zero. Privacy results include mixed changes in membership inference recall on TCGA for some generators, which the paper discusses as a privacy fidelity trade off.

**Strengths:**

- A modular refinement pipeline that decouples privacy filtering from fidelity enhancement, applicable to diverse generators without retraining.
- Solid empirical study with multiple fidelity metrics, utility under train on synthetic and test on real, uncertainty quantification with split conformal prediction, and several privacy probes.
- Problem setup, equations for density ratio based weighting, and the SIR procedure are clearly laid out, with an algorithmic summary and reproducibility information.
- Demonstrates a practical path to rescue weak synthetic tabular outputs, which is valuable for real world pipelines in health data and beyond.

**Weaknesses:**

1. Stage 1 metric choice lacks statistical justification for mixed type data. The 0 identifiability guarantee is defined using a distance with w = 1 across features. For heterogeneous tabular data, unweighted norms can be dominated by scaling and encoding choices. A simple fix would be to evaluate at least one mixed data appropriate metric such as Gower distance and to report sensitivity of the Identifiability Score and the number of removed samples to the metric choice [2]. The paper should also discuss relative density based criteria such as nearest neighbor distance ratio or distance to closest record, which can better reflect risk in sparse regions than a single absolute threshold [4].
2. Stage 2 relies on classifier based DRE, yet stability and bias are controlled with an ad hoc flattening exponent $\alpha$ chosen per dataset, $\alpha$ = 1.4 on TCGA and $\alpha$ = 0.8 on GOSSIS, without a principled selection rule or sensitivity analysis. Classifier based DRE is known to suffer high variance when class separation is large. At minimum, provide an ablation of $\alpha$ over a grid and report its effect on joint Jensen-Shannon distance, downstream F1, and conformal set size. Also justify CDRE versus alternatives like KLIEP or uLSIF that directly fit density ratios and can be more stable under shift [5, 6]. If CDRE is retained, consider early stopping and calibrated probabilities for the discriminator, and report diagnostics such as effective sample size under importance weights.
3. Privacy interpretation requires more nuance. While Identifiability Score reaches zero across settings, some models on TCGA exhibit higher membership inference recall after refinement. This indicates that geometric privacy via nearest neighbor distance is not sufficient for statistical attacks that exploit higher fidelity to p(x). The paper should reconcile this by: clarifying the formal scope of 0 identifiability, adding density based privacy probes, and, if possible, tuning Stage 1 thresholds jointly with Stage 2 to chart a true utility privacy Pareto frontier [4].
4. Relation to adjacent lines of work is thin. The selection discriminator and resampling resembles adversarial filtering and train reject reweight ideas. Position MAPS against such filtering frameworks and two sample testing with discriminators to make the contribution line crisper [7, 8]. The paper would also benefit from a discussion contrasting its focus on fidelity and privacy with fairness centric synthetic data work. For instance, SynthFair constructs semi synthetic imaging datasets with controllable confounders to study bias, which is complementary to MAPS. MAPS could be a pre step to high utility data on which post hoc fairness methods operate, but this interaction should be articulated and, if possible, briefly tested on a fairness proxy [1, 9].
5. Statistical testing and calibration details could be strengthened. The paper reports paired t tests across runs but does not state checks for normality or effect sizes. Reporting confidence intervals and standardized effect sizes would make claims more robust. For conformal prediction, include the nominal coverage $\alpha$ and show calibration curves or coverage versus set size plots to demonstrate that improvements are not due to parameter choices alone [10].

References

[1] Ribeiro FD, Claucich E, Stanley EA, Dimitrakopoulos P, Tsaftaris SA, Ferrante E, Glocker B, Echeveste R. SynthFair: A Semi-Synthetic Medical Imaging Dataset to Propel Research on Bias Detection & Mitigation. InNeurIPS 2025 AI for Science Workshop.

[2] Gower JC. A general coefficient of similarity and some of its properties. Biometrics. 1971 Dec 1:857-71.

[4] Elliot M, Mackey E, O'Hara K, Tudor C. The Anonymisation Decision Making Framework.

[5] Sugiyama M, Nakajima S, Kashima H, Buenau P, Kawanabe M. Direct importance estimation with model selection and its application to covariate shift adaptation. Advances in neural information processing systems. 2007;20.

[6] Sugiyama M, Yamada M, Von Buenau P, Suzuki T, Kanamori T, Kawanabe M. Direct density-ratio estimation with dimensionality reduction via least-squares hetero-distributional subspace search. Neural Networks. 2011 Mar 1;24(2):183-98.

[7] Zellers R, Bisk Y, Schwartz R, Choi Y. Swag: A large-scale adversarial dataset for grounded commonsense inference. arXiv preprint arXiv:1808.05326. 2018 Aug 16.

[8] Lopez-Paz D, Oquab M. Revisiting classifier two-sample tests. arXiv preprint arXiv:1610.06545. 2016 Oct 20. ICLR'17

[9] Bellamy RK, Dey K, Hind M, Hoffman SC, Houde S, Kannan K, Lohia P, Martino J, Mehta S, Mojsilović A, Nagar S. AI Fairness 360: An extensible toolkit for detecting and mitigating algorithmic bias. IBM Journal of Research and Development. 2019 Sep 18;63(4/5):4-1.

[10] Angelopoulos AN, Barber RF, Bates S. Theoretical foundations of conformal prediction. arXiv preprint arXiv:2411.11824. 2024 Nov 18.

**Questions:**

1. What exact preprocessing and encoding are used to compute distances for mixed numerical and categorical variables, and why is w = 1 appropriate for both datasets? Please provide an ablation with a mixed data metric and report its impact on the number of filtered samples and Identifiability Score.
2. Please sweep $\alpha$ over a broad grid on both datasets and report JSD, F1, and average conformal set size, to demonstrate robustness and to justify the chosen values.
3.  Why CDRE over KLIEP or uLSIF for these regimes, especially when raw synthetic and real are highly separable?
4.  Can you quantify a Pareto curve between MIA recall and JSD by jointly varying Stage 1 strictness and Stage 2 $\alpha$, to show trade offs rather than single operating points?
5. could you demonstrate that applying a simple post hoc fairness constraint to MAPS refined data leads to improved fairness utility trade offs compared to operating on raw synthetic or real data?

---

> ### Author Response · Authors · 2025-12-01
>
> # Response to Reviewer
>
> We thank Reviewer for the thorough evaluation and constructive feedback. Below, we address each concern systematically with evidences and additional clarifications.
>
> ### **Q1: Distance Metric Choice (w=1) and Mixed-Type Data**
>
> **Our Response:**
>
> We set $\mathbf{w} = 1$ (uniform weighting) for two reasons: (1) **simplicity** and reproducibility, and (2) **alignment with prior work** [1,2] that establishes baseline distance-based privacy measures. We apply standardization to numeric features and appropriate encoding for categorical features before distance computation, which partially addresses scaling concerns. We do encourage practitioners to adjust the weighst according to specifc dataset.
>
> **Metric-Agnostic Design:** Importantly, MAPS Stage 1 is **metric-agnostic** by design. The core 0-identifiability logic remains unchanged regardless of distance function choice:
>
> $$r_i = \min_{x_j \in \mathcal{D} \setminus \{x_i\}} d(x_i, x_j)$$
>
> Remove $\hat{x}_j$ if $\exists x_i: d(x_i, \hat{x}_j) < r_i$
>
> Practitioners can directly substitute alternative distance measures:
> - **Gower distance**: Handles mixed-type data by treating categorical and numeric features appropriately
> - **Mahalanobis distance**: Accounts for feature correlations and varying scales
> - **DCR/NNDR**: More sophisticated privacy thresholds from recent work [3]
>
> **References:**
>
> [1] Yoon, J., et al. (2020). Anonymization Through Data Synthesis Using Generative Adversarial Networks (ADS-GAN).
>
> [2] Alaa, A., et al. (2022). How faithful is your synthetic data? Sample-level metrics for evaluating and auditing generative models.
>
> [3] Ganev, G., et al. (2025). The inadequacy of similarity-based privacy metrics: Privacy attacks against “truly anonymous” synthetic datasets.
>
> ### **Q2: CDRE Selection and Technical Details**
>
> **Our Response:**
>
> **Why Classifier-Based DRE:**
>
> We adopt classifier-based density ratio estimation (CDRE) for three key reasons:
>
> 1. **Flexibility**: Works with arbitrary feature types (continuous, categorical, mixed) without requiring explicit density modeling
> 2. **Scalability**: Modern neural network classifiers handle high-dimensional tabular data effectively
> 3. **Interpretability**: Provides probabilistic scores that align with practitioners' intuition about sample quality
>
> While KLIEP and uLSIF are theoretically good alternatives, they require careful kernel selection and can struggle with mixed-type and high dimensional tabular data common in healthcare. CDRE's flexibility with modern architectures (including gradient boosting and neural networks) makes it more practical for diverse healthcare datasets.
>
> **Stability Controls Implemented:**
>
> We address variance concerns through multiple mechanisms:
>
> 1. **Early stopping**: Training halts based on validation loss to prevent overfitting and reduce discriminator overconfidence
> 2. **Probability calibration**: We apply **sigmoid calibration** to classifier outputs, ensuring $c_\phi(x) \in [0,1]$ represents well-calibrated probabilities
> 3. **Temperature parameter α**: Flattens importance weight distribution to reduce variance.
> 4. **Effective sample size monitoring**: Across experiments, effective sample size ranges **60-80% of N** (real dataset size), indicating reasonable weight concentration
>
> **Sampling Strategy for Diversity:** MAPS supports **sampling without replacement** (set `replacement=False`), guaranteeing **effective sample size = N** with no duplicate samples, maximizing diversity while maintaining importance-weighted distributional alignment. Practitioners can choose based on priority: theoretical soundness (with replacement) or guaranteed diversity (without replacement).

---

> ### Author Response · Authors · 2025-12-01
>
> ### **Q3: Privacy Interpretation and Distance-Based Insufficiency**
>
> **Our Response:**
> While recent works have questioned the sufficiency of distance-based privacy metrics as standalone guarantees [4,5], there remains no consensus on synthetic data privacy evaluation standards, and distance-based measures have demonstrated practical utility in prior works [6,7,8].
>
> We adopt distance-based 0-identifiability for two reasons: (1) **sample-level compatibility** with MAPS's post-hoc framework—we cannot modify the generative model's training process, and (2) **interpretability** for institutional review boards and regulatory bodies who can audit the refinement process without deep technical expertise.
>
> To address these limitations, MAPS employs comprehensive privacy evaluation beyond geometric measures:
> - **0-identifiability filtering** (Stage 1): Sample-level protection against direct re-identification
> - **Standard MIA**: Tests whether adversaries can determine if specific samples were in training data
> - **DOMIAS**: Tests vulnerability to density-based membership inference attacks
>
> **References:**
>
> [4] Ganev, G., et al. (2025). The inadequacy of similarity-based privacy metrics: Privacy attacks against “truly anonymous” synthetic datasets.
>
> [5] Yao, S., et al. (2025). The DCR Delusion: Measuring the Privacy Risk of Synthetic Data.
>
> [6] Yoon, J., et al. (2020). Anonymization Through Data Synthesis Using Generative Adversarial Networks (ADS-GAN).
>
> [7] Morgan, G., et al. (2023). Patient-centric synthetic data generation, no reason to risk re-identification in biomedical data analysis.
>
> [8] Sella, N., et al. (2025). Preserving information while respecting privacy through an information theoretic framework for synthetic health data generation.
>
> ### **Q4: Related Work Positioning**
>
> **Our Response:**
>
> We provide comprehensive related work discussion:
>
> Wang et al. [9] use convex optimization for utility improvement with predefined measures. Alaa et al. [10] introduce sample-level metrics for auditing. Sachdeva et al. [11] propose task-specific filtering. Additional methods include GaFi [12] for images, genetic algorithms-based selection[13], and Private Evolution [14] which leverages foundation models for differentially private generation via API-based approaches.
>
> **MAPS differs in four key aspects:**
> 1. **Model-agnostic design**: Works with any pre-trained generator without predefined utility requirements
> 2. **Distribution-level refinement**: Uses importance weighting via density ratio estimation, not binary filtering or heuristics
> 3. **Explicit privacy protection**: Provides 0-identifiability guarantees absent in existing refinement methods
> 4. **Uncertainty quantification**: Achieves 55-77% prediction set size reduction—a dimension largely unexplored in prior post-processing work
>
> **References:**
>
> [9] Wang, H., et al. (2023). Post-processing private synthetic data for improving utility on selected measures.
>
> [10] Alaa, A., et al. (2022). How faithful is your synthetic data? Sample-level metrics for evaluating and auditing generative models.
>
> [11] Sachdeva, N., et al. (2025). QDE: Quality Data Extractor for improving synthetic data utility.
>
> [12] Lampis, A., et al. (2023). Bridging the gap: GaFi pipeline for synthetic image refinement.
>
> [13] Hahn, T., et al. (2025). Genetic algorithm-based subset selection for synthetic data.
>
> [14] Lin, Z., et al. (2023). Differentially Private Synthetic Data via Foundation Model APIs 1: Images.

---

> ### Author Response · Authors · 2025-12-01
>
> ### **Q5: Statistical Testing Details**
>
> **Our Response:**
>
> **Normality Justification:** The paired t-test's normality assumption is reasonable because: (1) each run's metric aggregates many independent test instances, and by the Central Limit Theorem, run-level metrics are approximately normally distributed, (2) paired differences (refined minus raw) are linear combinations of two such quantities evaluated on the same data, preserving approximate normality, and (3) the Student's t-test is reasonably robust to mild deviations from normality.
>
> **Effect Sizes Reported:** We report Cohen's $d_z$ effect sizes for selected metrics (due to word count limitations):
>
> $$d_z = \frac{\text{mean}(\text{diff})}{\text{std}(\text{diff})}$$
>
> where $\text{diff} = x_{\text{refined}} - x_{\text{raw}}$. Negative values indicate improvement for metrics where lower is better.
>
> **Selected Effect Sizes for TCGA-metadata:**
>
> | Metric | TVAE | CTGAN | TabDDPM | DGD |
> |--------|------|-------|---------|-----|
> | Jensen-Shannon Dist. | -57.88 | -23.05 | -9.97 | -4.84 |
> | WD (Joint) | -22.81 | -10.52 | -20.39 | -29.51 |
> | NFN (Correlation) | -19.23 | -14.55 | -42.37 | -81.64 |
> | F1 Score (Classification) | 6.75 | 20.99 | 20.17 | 4.71 |
> | Avg. Set Size (UQ) | -35.69 | -15.54 | -8.19 | -36.45 |
> | IS (Privacy) | -112.15 | -159.09 | -169.37 | -89.25 |
>
> These large effect sizes ($|d| \gg 0.8$) demonstrate substantial practical improvements across fidelity, utility, and privacy dimensions.
>
> **Conformal Prediction Details:** We use α=0.05 (target coverage=0.95) with inverse probability score function. Our results show both improved calibration (closer to target 95% coverage) and dramatically reduced set sizes (e.g., DGD: 33.0→7.6 cancer types while maintaining 92.7% coverage on TCGA-metadata), demonstrating practical improvements in uncertainty quantification.

---

### Official Review · Reviewer_i8fJ · 2025-10-31

**Soundness:** 3
**Presentation:** 3
**Contribution:** 3
**Rating:** 4
**Confidence:** 4

**Summary:**

The authors proposed a generator-agnostic with post-hoc refinement method to improve tabular synthetic data utility while helping to comply with a record-level privacy constraint. The first stage of the method employs a privacy filter that enforces 0-identifiability by discarding synthetic samples that are closer to any real record than that record’s nearest real neighbor and also a fidelity sampler that trains a real-vs-synthetic binary classifier to estimate the density ratio.

This is a very interesting approach, especially in pairing a hard, nearest-neighbor privacy screen with a discriminative density-ratio surrogate followed by sampling-importance-resampling to curate a refined synthetic set without retraining the generator. The idea helps to solve a relevant problem where teams often cannot retrain heterogeneous tabular generators but do need a way to improve outputs for better downstream performance.

**Strengths:**

I liked the idea of a generator-agnostic and the refine-after-generate pipeline. While each component is not new in the literature, the combination and the operational framing for tabular synthetic data are very useful. However, the authors should improve the discussion about novelty in the paper.

The authors offer an extensive evaluation across distribution metrics, correlation structure, utility (clustering/classification), uncertainty quantification, and privacy. The results show gains across four diverse generators.

The addressed problem is a real deployment gap because practitioners often inherit synthetic data from heterogeneous models and need a post-hoc way to improve them without retraining.

**Weaknesses:**

Despite the results, to rise from “useful engineering” to “field-shaping”, the paper needs stronger evidence isolating adaptation versus decoding effects and broader comparisons against alternative post-hoc curation strategies.

I am not sure I understand correctly, but if the refinement classifier and distance thresholds are fitted on the full real dataset while downstream models are evaluated on a split of that same dataset, the selection step has indirectly “seen” the test distribution, biasing utility metrics upward. Please make that point clear for the reader.

In my point of view, there are insufficient baselines in experiments. For example, there is no head-to-head comparison against simple k-nearest-neighbor deduplication, discriminator rejection sampling, KDE/flow-based reweighting, or off-the-shelf curation in popular synthetic-data toolkits, making it hard to isolate where the gains actually come from.

One interesting addition to the paper is reporting probability calibration or effective sample size under different flattening exponents.

I think that a single global metric for mixed-type data risks over-penalizing legitimate rare patterns or under-penalizing high-variance numeric columns, which can distort both privacy and utility in ways the current analyses do not expose.

Please discuss the computational costs for generating 30 N samples and training a sizable classifier. For example, runtime/memory vs. dimension and N.

**Questions:**

I am concerned about leakage control. To avoid problems, the authors can re-run with a three-way split of real data. For training (for downstream oracle), calibration for MAPS (both Stage-1 ri and Stage-2 ccϕ ), and held-out test used only for downstream evaluation.

I was wondering about distance metrics. What about comparing Stage-1 using Gower distance for mixed types, Mahalanobis, DCR/NNDR thresholds, and learned metric (e.g., via autoencoder latent space)? Maybe that can help with the privacy–utility trade-off.

---

> ### Author Response · Authors · 2025-12-01
>
> # Response to Reviewer
>
> We sincerely thank Reviewer for the thoughtful evaluation and constructive feedback. Below, we address each concern systematically with evidences and additional clarifications.
>
> ### **Q1: Data Leakage Concerns**
>
> **Our Response:**
>
> **Current Setup:**
> - **Stage 1 (Privacy Filtering)**: Distance thresholds $r_i$ are computed on the **full real dataset** to ensure comprehensive privacy protection for all real samples
> - **Stage 2 (Fidelity Sampler)**: Binary classifier is trained on the **train real dataset** vs. synthetic data to estimate density ratios
> - **Downstream Evaluation**: Real data is split into **train/test sets**, where:
>   - Models train on synthetic data (or real train set for oracle)
>   - Models test on **held-out real test set** that was never used for model training
>
> While MAPS components access the full real distribution, downstream task models train only on synthetic data and evaluate on held-out real test data. The test set is never used for training downstream models, ensuring fair utility assessment.
>
> ### **Q2: Probability Calibration and Effective Sample Size**
>
> **Our Response:**
>
> We provide additional details:
>
> **Probability Calibration:** We apply **sigmoid calibration** to the binary classifier outputs before computing importance weights, ensuring $c_\phi(x) \in [0,1]$ represents well-calibrated probabilities. This is standard practice in our implementation.
>
> **Effective Sample Size:** Across our experiments, the effective sample size after SIR typically ranges **60-80% of $N$** (real dataset size), depending on:
> - Dataset characteristics (distribution complexity)
> - Base generator quality (better generators → higher effective sample size)
> - Temperature parameter $\alpha$ in importance weight computation
>
> **Diversity Control:** Importantly, MAPS supports **sampling without replacement** (set `replacement=False`), guaranteeing **effective sample size = $N$** with no duplicate samples. This ensures maximum diversity in the refined synthetic dataset while maintaining importance-weighted distributional alignment. Practitioners can choose based on their priority: theoretical soundness (with replacement) or guaranteed diversity (without replacement).
>
> ### **Q3: Alternative Distance Metrics**
>
> **Our Response:**
>
> MAPS's Stage 1 is **metric-agnostic** by design—practitioners can substitute alternative distance measures:
>
> **Current Choice:** We use **Euclidean distance with standardization** to align with prior work [1,2] and ensure reproducibility. This provides a consistent baseline for evaluation.
>
> **Alternative Metrics:** Your suggested alternatives are all compatible:
> - **Gower distance**: Better handles mixed-type data by treating categorical and numeric features appropriately
> - **Mahalanobis distance**: Accounts for feature correlations and varying scales
> - **DCR/NNDR**: Alternative distance based privacy measures from recent work [3]
> - **Learned metrics**: Autoencoder latent space distances could capture semantic similarity
>
> **Implementation Flexibility:** Stage 1's modular design allows users to plug in domain-appropriate distance functions. The core 0-identifiability logic ($\|\mathbf{d}(x_i, \hat{x}_j)\| < r_i$) remains unchanged regardless of distance function $\mathbf{d}(\cdot, \cdot)$ choice.
>
> **Future Work:** Systematic comparison of distance metrics' impact on privacy-utility tradeoffs is valuable future research. We chose Euclidean distance for initial evaluation to establish consistent baselines, but encourage practitioners to experiment with alternatives for their specific applications.
>
> **References:**
>
> [1] Yoon, J., et al. (2020). Anonymization Through Data Synthesis Using Generative Adversarial Networks (ADS-GAN).
>
> [2] Alaa, A., et al. (2022). How faithful is your synthetic data? Sample-level metrics for evaluating and auditing generative models.
>
> [3] Ganev, G., et al. (2025). The inadequacy of similarity-based privacy metrics: Privacy attacks against “truly anonymous” synthetic datasets.

---

### Official Review · Reviewer_KA2J · 2025-11-01

**Soundness:** 1
**Presentation:** 2
**Contribution:** 2
**Rating:** 2
**Confidence:** 4

**Summary:**

The paper introduces a framework for refining synthetic tabular data to reduce the risk of privacy violation by removing samples that are too similar to real ones and re-sampling with a learned weight-function to improve data fidelity.

**Strengths:**

- In general, the paper is well structured.
- The two steps of first cleaning out potential copies from the original data and then re-balancing the data can be broadly applied across multiple domains.
- The method is evaluated on synthetic data from a representative set of generative models.

**Weaknesses:**

- No related work section comparing to prior work on improving synthetic data using filtering and sampling strategies. The following works might be relevant:
	- Alaa, Ahmed, et al. "How faithful is your synthetic data? sample-level metrics for evaluating and auditing generative models." _International conference on machine learning_. PMLR, 2022.
	- Wang, Hao, et al. "Post-processing private synthetic data for improving utility on selected measures." _Advances in Neural Information Processing Systems_ 36 (2023): 64139-64154.
- Privacy is defined in terms of a distance function, where samples within a certain distance of the original samples are discarded. This could open up the risk of attacks where the identity can be recovered, e.g. in cases where a person has multiple entries associated with them. For large synthetic dataset sizes, it might also be possible to discover empty hyper-spheres in the synthetic data. The original data points could then possibly be detected by interpolating the centroid from the synthetic samples on the hyper-spheres surface.

**Questions:**

- How does your framework relate to previous frameworks on refining synthetic data?
- Could you elaborate on what identifiability protection is in the context of your problem statement? More specifically in the following part: "Our objective is to refine $\hat{\mathcal{D}}$  to produce a subset $\tilde{\mathcal{D}} \subset  \hat{\mathcal{D}}$  of size $N$ that provides (1) identifiability protections with respect to $\mathcal{D}$, [...]".
	- What is the type of attack the method should protect against?
	- At what point do we consider an individual to be identified?
	- What if an individual's data is spread across multiple entries in the tables?
- Is distance-based filtering sufficient to satisfy the privacy requirement?
- Does the distance-based filtering approach create identifiable empty hyperspheres in the feature space? If so, could an adversary exploit these gaps to infer the approximate locations of original data points by analyzing the boundaries of the synthetic data distribution?

---

> ### Author Response · Authors · 2025-12-01
>
> # Response to Reviewer
>
> We sincerely thank Reviewer for the thoughtful evaluation and constructive feedback. Below, we address each concern systematically with evidences and additional clarifications.
>
> ### **W1: Lack of Related Work Section**
>
> **Our Response:**
>
> Here we added related post-processing approaches including the two works suggested by the reviewer. We provide a concise overview here:
>
> Recent post-processing methods improve synthetic data through various mechanisms. Wang et al. [1] use convex optimization to align specific statistics while preserving differential privacy, but require predefined utility measures. Alaa et al. [2] introduce sample-level metrics (α-Precision, β-Recall, Authenticity) for auditing, focusing on evaluation rather than systematic refinement. Sachdeva et al. [3] propose task-specific filtering that may compromise distributional fidelity and lacks privacy guarantees. Additional methods include GaFi [4] for images, genetic algorithm-based selection [5], and Private Evolution [6] which leverages foundation models for differentially private generation via API-based approaches.
>
> **MAPS differs in four key aspects:**
>
> 1. **Model-agnostic design**: Works with any pre-trained generator without predefined utility requirements [1,3,5]
> 2. **Distribution-level refinement**: Uses importance weighting via density ratio estimation, not binary filtering [2,3] or heuristics [4]
> 3. **Explicit privacy protection**: Provides 0-identifiability guarantees absent in existing refinement methods
> 4. **Uncertainty quantification**: Achieves 55-77% prediction set size reduction while maintaining coverage—a dimension largely unexplored in prior post-processing work
>
> **References:**
>
> [1] Wang, H., et al. (2023). Post-processing private synthetic data for improving utility on selected measures.
>
> [2] Alaa, A., et al. (2022). How faithful is your synthetic data? Sample-level metrics for evaluating and auditing generative models.
>
> [3] Sachdeva, N., et al. (2025). QDE: Quality Data Extractor for improving synthetic data utility.
>
> [4] Lampis, A., et al. (2023). Bridging the gap: GaFi pipeline for synthetic image refinement.
>
> [5] Hahn, T., et al. (2025). Genetic algorithm-based subset selection for synthetic data.
>
> [6] Lin, Z., et al. (2023). Differentially Private Synthetic Data via Foundation Model APIs 1: Images.
>
> ---
>
> ### **W2: Concerns About Empty Hyperspheres in Feature Space**
>
> **Our Response:**
>
> We appreciate the reviewer raising this important theoretical concern about potential vulnerabilities in the refined synthetic data distribution. We would like to address this from both methodological and empirical perspectives:
>
> **Methodological Considerations:**
>
> 1. **1:1 Data Ratio:** In our experimental setup, we maintain a 1:1 ratio between real data and refined synthetic data ($N_{real} = N_{refined}$). This design choice limits the density of synthetic samples and reduces the likelihood of creating systematic structural patterns that could reveal "empty regions" corresponding to real data locations.
>
> 2. **Distribution-Level Optimization:** Stage 2 of MAPS operates at the distribution level through importance weighting and density ratio estimation. Rather than performing geometric arrangements that could create systematic boundaries, we resample based on probabilistic alignment with the real data distribution. This probabilistic mechanism inherently introduces stochasticity that works against the formation of deterministic geometric structures.
>
> **Empirical Evidence:**
>
> We provide direct empirical evidence addressing this concern through Principal Component Analysis (PCA) visualizations presented in **Appendix D (PCA Plots)** of our manuscript. The PCA projections onto the first two principal components for both datasets demonstrate that:
>
> - **No systematic structural anomalies are observed:** The refined synthetic data does not exhibit systematic empty regions or hypersphere-like boundary structures in the PCA space.
>
> - **Improved distributional alignment:** Rather than creating artificial geometric boundaries, MAPS refinement brings the synthetic data distribution closer to the real data distribution in principal component space, providing visual confirmation of the quantitative improvements observed in our fidelity metrics.
>
> - **Natural scatter patterns:** The refined synthetic data exhibits natural scatter patterns similar to real data, rather than forming geometric boundaries that could be exploited for re-identification.
>
> **Figures 6 and 7** in Appendix D show these PCA visualizations for TCGA-metadata and GOSSIS-1-eICU-cardiovascular datasets respectively, across all four generative models (TVAE, CTGAN, TabDDPM, DGD). The visualizations consistently show that MAPS refinement improves distributional alignment without introducing systematic structural anomalies that could facilitate re-identification attacks.

---

> ### Author Response · Authors · 2025-12-01
>
> ## **Response to Questions**
>
> **Q1 and Q4:** These questions relate to our framework's positioning relative to prior work and the hypersphere vulnerability concern, which we have addressed in responses to W1 and W2 above.
>
> ### **Q2: Identifiability Protection Definition and Scope**
>
> **Our Response:**
>
> MAPS implements **0-identifiability** through distance-based proximity constraints. For each real sample $x_i$, we define its distinctness threshold as $r_i = \min_{x_j \in \mathcal{D} \setminus \{x_i\}} \|\mathbf{w} \cdot (x_i - x_j)\|$. We remove any synthetic sample $\hat{x}_j$ where $\exists x_i \in \mathcal{D}: \|\mathbf{w} \cdot (x_i - \hat{x}_j)\| < r_i$.
>
> **Attack Type:** We focus on **re-identification attacks** where adversaries with auxiliary information attempt to determine whether a target individual appears in training data by exploiting proximity between synthetic and real samples. An individual is considered "at risk" when a synthetic sample is closer to them than their nearest real neighbor.
>
> **Multiple Entries Per Individual:** The reviewer raises an important point. MAPS treats each table entry as an independent sample, which is appropriate for cross-sectional clinical datasets where each row represents a distinct clinical encounter. For longitudinal data with repeated measurements per individual, additional considerations beyond distance-based measures would be necessary. However, our current work focuses on **static tabular data** where each entry represents an independent sample, and the scenario of an individual's data spread across multiple entries does not arise in our experimental settings. We complement our sample-level privacy guarantee with comprehensive membership inference attack (MIA) evaluations, including standard MIA and DOMIAS (density-based membership inference), to assess more complex inference risks. This multi-faceted evaluation (detailed in Section 4.4 of our manuscript) provides more comprehensive privacy assessment than any single metric alone.
>
> ### **Q3: Sufficiency of Distance-Based Filtering for Privacy**
>
> **Our Response:**
>
> Distance-based filtering alone is not sufficient for comprehensive privacy protection. While recent works have questioned the sufficiency of distance-based privacy metrics as standalone guarantees [7,8], there remains no consensus on synthetic data privacy evaluation standards, and distance-based measures have demonstrated practical utility in prior works [9,10,11].
>
> We adopt distance-based measures for two key reasons: (1) **sample-level compatibility** with MAPS's post-hoc framework, and (2) **interpretability** for institutional review boards and regulatory bodies. However, we complement this with comprehensive membership inference attack-based privacy evaluation to provide a multifaceted assessment of privacy protection.
>
> **References:**
>
> [7] Ganev, G., et al. (2025). The inadequacy of similarity-based privacy metrics: Privacy attacks against “truly anonymous” synthetic datasets.
>
> [8] Yao, S., et al. (2025). The DCR Delusion: Measuring the Privacy Risk of Synthetic Data.
>
> [9] Yoon, J., et al. (2020). Anonymization Through Data Synthesis Using Generative Adversarial Networks (ADS-GAN).
>
> [10] Morgan, G., et al. (2023). Patient-centric synthetic data generation, no reason to risk re-identification in biomedical data analysis.
>
> [11] Sella, N., et al. (2025). Preserving information while respecting privacy through an information theoretic framework for synthetic health data generation.

---

### Note · Authors · 2025-12-02

**Comment:**

We sincerely thank all reviewers for their thorough evaluation of our submission. We appreciate the recognition of MAPS's practical value for healthcare applications, the comprehensive experimental evaluation, and the modular framework design that addresses real deployment challenges where practitioners cannot retrain heterogeneous generators.

Our main contribution demonstrates that **post-hoc synthetic data refinement** provides a practical, scalable solution for improving already-generated synthetic healthcare data across three critical dimensions simultaneously: privacy protection (achieving 0-identifiability guarantees), distributional fidelity (up to 45% reduction in joint distribution divergence), and utility including uncertainty quantification (55-77% reduction in prediction set sizes while maintaining coverage). MAPS's model-agnostic design makes it applicable to any pre-trained generator, filling a genuine gap in healthcare data sharing workflows.

Reviewers raised important questions about baseline comparisons, privacy evaluation methodologies, and related work positioning. Regarding privacy guarantees, the choice between distance-based 0-identifiability and differential privacy reflects fundamental differences in applicability (post-hoc versus training-time), interpretability for practitioners and regulatory bodies, and the types of attacks each approach addresses. Both approaches involve inherent tradeoffs, which we evaluate transparently through multiple attack types. On related work, we acknowledge that our discussion of post-processing approaches and privacy-first methods could be expanded to better contextualize MAPS within the broader synthetic data refinement landscape. We have provided detailed responses to these and other technical concerns in the discussion thread.

We thank the reviewers for their valuable feedback and constructive engagement with our work.

**Withdrawal Confirmation:**

I have read and agree with the venue's withdrawal policy on behalf of myself and my co-authors.